# Seabirds shaped the expansion of pre-Inca society in Peru

**Jacob L. Bongers** [1,2,3]*, **Emily B. P. Milton** [4]*, **Jo Osborn** [5], **Dorothée G. Drucker** [6], **Joshua R. Robinson** [7], **Beth K. Scaffidi** [8]

1 Discipline of Archaeology, School of Humanities, Faculty of Arts and Social Sciences, The University of Sydney, Sydney, New South Wales, Australia, 2 The Vere Gordon Childe Centre, Faculty of Arts and Social Sciences, The University of Sydney, Sydney, New South Wales, Australia, 3 Australian Museum Research Institute, Australian Museum, Sydney, New South Wales, Australia, 4 Department of Anthropology, National Museum of Natural History, Smithsonian Institution, Washington, DC, United States of America, 5 Department of Anthropology, Texas A&M University, College Station, Texas, United States of America, 6 Senckenberg Centre for Human Evolution and Palaeoenvironment, Department of Geosciences, University of Tübingen, Tübingen, Germany, 7 Archaeology Program, Boston University, Boston, Massachusetts, United States of America, 8 Department of Anthropology and Heritage Studies, University of California, Merced, United States of America

* jacob.bongers@sydney.edu.au (JLB); Miltone@si.edu (EBPM)

## Abstract

This research investigates the influence of seabird guano on agriculture in the Chincha Valley of southern Peru through multi-isotopic, archaeological, and historical data. We conduct stable carbon, nitrogen, and sulfur analyses of 35 late pre-Hispanic maize (*Zea mays*) cobs and 11 seabirds from archaeological contexts spanning the late Formative period (*c.* 200 BCE – 150 CE) to the Colonial period (1532–1825 CE). We report the strongest evidence yet for pre-Inca use of marine fertilizers in Chincha. Isotopic and radiocarbon data corroborate colonial-era records and regional avifauna iconography and assemblages, indicating that Indigenous communities fertilized maize with guano by at least 1250 CE. Maize $\delta^{15}N$ values are consistent with archaeological studies on guano manuring in Chile, expanding the known geographical extent of this agricultural practice. Maize $\delta^{34}S$ values overlap with experimental field data but are not enriched in $^{34}S$, possibly reflecting various environmental and cultural variables. We suggest that seabird guano fertilization played an important role in the sociopolitical and economic expansion of the Chincha Kingdom, and its eventual relationship with the Inca Empire. Our findings carry significant implications for the broader Andes, nuancing understandings of agricultural production in coastal environments while drawing attention to marine fertilizers as a potentially widespread driving force of social change among pre-Hispanic societies.

**Data availability statement:** All relevant data are within the paper and its Supporting information files.

**Funding:** Funding for archaeological field-work and isotopic analyses of maize samples was provided to JLB by the National Science Foundation Graduate Research Fellowship Program (DGE-1144087), the Society of Fellows at Boston University, the Ford Foundation Fellowship Program, the National Geographic Young Explorers Grant Program (9347-13), and the Sigma Xi Grants-in-Aid Research Program. The funders had no role in the study design, data collection and analysis, decision to publish, or preparation of the manuscript.

**Competing interests:** The authors have declared that no competing interests exist.

## Introduction

Maize (*Zea mays*), a globally significant staple grain [1], had become a dominant crop across the Americas by the onset of European colonialism. However, the mechanisms supporting its geographic expansion and diverse domestication remain unknown [2]. Animal management likely contributed to these processes, given its long-standing role in supporting human subsistence [3]. For example, manuring with animal waste is an effective and sustainable practice because it enhances crop yields and maintains soil fertility [4], which in turn can contribute to the expansion and persistence of societies by ensuring the continuous provision of food for growing populations [5,6]. In the 19th century, guano—seabird excrement and accumulated food waste, feathers, and carcasses—from "guano islands" off the western coast of South America became one of the world's most sought-after fertilizers because it contains the essential growing nutrients of nitrogen, phosphorous, and potassium (NPK) [7–9]. Recent research suggests that guano fertilization may have begun by at least 1000 CE in Tarapacá, northern Chile, yet the origins and regional importance of this fertilizer are poorly understood [10]. Using archaeological, historical, and isotopic data from the Chincha Valley, Peru, we ask: to what extent did seabird guano shape the development of pre-Hispanic societies in the Andes?

Here we apply a multidisciplinary, multi-isotopic ($\delta^{13}$C, $\delta^{15}$N, and $\delta^{34}$S) approach to evaluate seabird guano manuring in the Chincha Valley on the Peruvian southern coast (Fig 1) and potential paleodietary isotopic complications caused by this practice. Chincha is one of the most agriculturally productive coastal riverine valleys in Peru, making it an ideal context for this study. It was controlled by the Chincha Kingdom, a large-scale and centralized polity, during the Late Intermediate Period (LIP; 1000–1400 CE) before falling to the Inca and Spanish empires in the Late Horizon (LH or Inca period; 1400–1532 CE) and Colonial period, respectively. This polity reportedly comprised at least 30,000 tribute payers, organized into inter-dependent communities of specialized farmers, fisherfolk, and merchants [12]. High population coincided with a dense distribution of LIP and LH habitational sites, administrative complexes, cemeteries, and farming infrastructure (e.g., irrigation canals and cultivation fields) throughout the Chincha Valley, suggesting that demographic growth and agricultural production both increased during these periods [13]. This coastal valley is within 25 kilometers of the Chincha Islands, which are renowned for their abundant and high-quality guano deposits [7]. Guano-producing birds include the guanay cormorant (*Leucocarbo bougainvilliorum*), the Peruvian booby (*Sula variegata*), and the Peruvian pelican (*Pelecanus thagus*) (Fig 2). Other birds, including penguins and gulls, can also contribute to guano accumulation [8]. The proximity of these historically important deposits suggests that pre-Hispanic farmers in Chincha likely utilized seabird guano as a fertilizer [14].

Stable nitrogen analysis provides a reliable method for detecting past manuring practices [8,10]. Archaeological investigations of manure fertilization in southern Peru and northern Chile have found elevated nitrogen stable isotope ratios ($\delta^{15}$N) in crops, especially maize, suggesting farmers fertilized their fields with guano prior to the Spanish Colonial period (1532–1825 CE) [10,15]. Experimental studies demonstrate

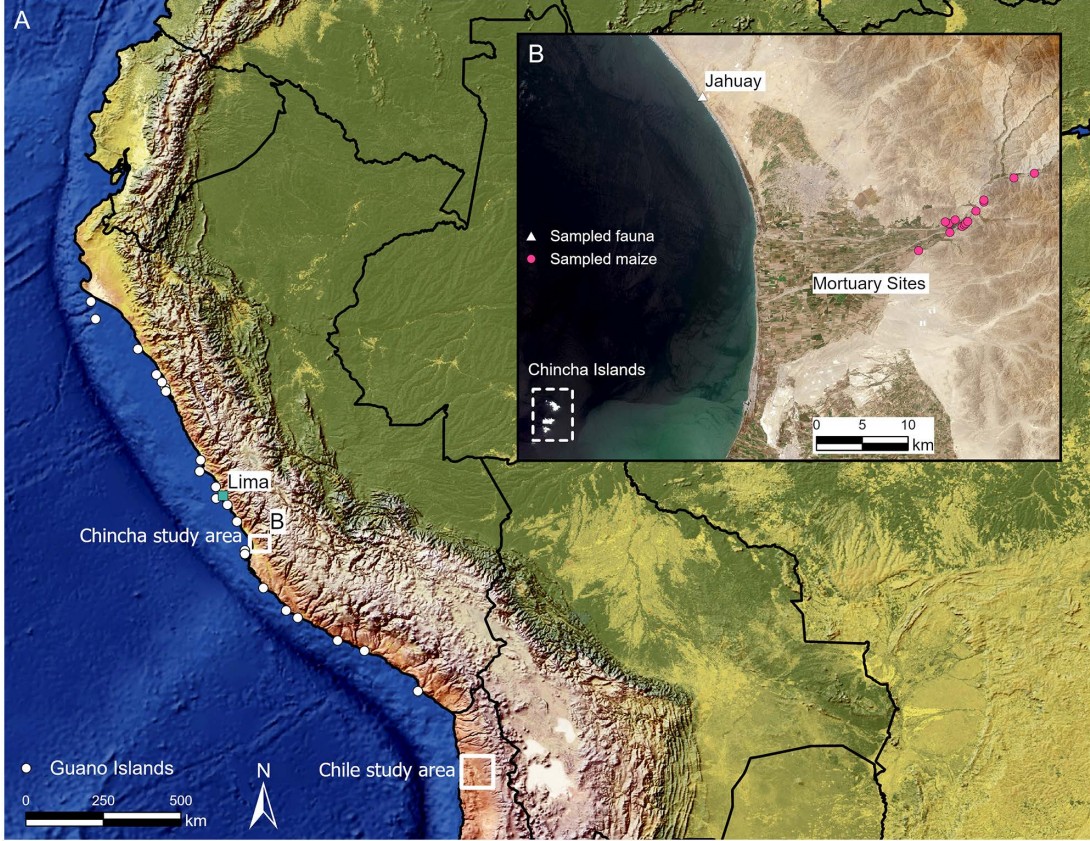

**Fig 1. Contextual information for maize and faunal isotopic data. (A)** Map of Peruvian guano islands [11] and Chincha and Chile [10] study areas, which yielded maize isotopic data that are compared in this study. **(B)** Map of Chincha study area, with middle valley cemeteries sampled for maize (marked by pink circles), the site of Jahuay (marked by white triangle) sampled for faunal remains, and the Chincha Islands (marked by the dotted white line). Shaded relief and country borders (A) are from Natural Earth. Sentinel-2 satellite imagery **(B)** was freely downloaded from https://dataspace.copernicus.eu/.

that crop nitrogen isotope ratios are altered by various biogenic manures, such as guano, fish remains, algae, and herbivore dung [8,16,17]. Organically fertilized plants tend to be more enriched in [15]N relative to unfertilized plants, with significant increases in plant $\delta^{15}$N values by as much as +2 to +10‰ for terrestrial manures [18], +2 to +15‰ for marine composts [17], and +10 to +40‰ for seabird guano [8]. When measured plant $\delta^{15}$N values are less than ~12–15‰, confirming the presence and type of fertilization practices can be complicated due to other nitrogen fractionation processes including intra-plant uptake variability [8], aridity [18], soil composition [19], and water management practices [20]. However, few natural processes and no marine fertilizers are known to produce $\delta^{15}$N values higher than +20‰, except seabird guano [16,21].

Recently, stable sulfur ($\delta^{34}$S) has been proposed as a secondary mechanism of identifying the use of marine fertilizers on crops [16,17]. Experimental studies have demonstrated that multiple marine fertilizers can result in significant [34]S enrichment (+4‰ to +9‰) in plants [16,17]; however, associated field studies provided inconclusive results for bean and maize cultivars, suggesting that local growing conditions and plant physiology may influence the uptake of guano-derived $\delta^{34}$S values [9]. The exact mechanisms of sulfur uptake and realistic regional expectations for $\delta^{34}$S values in guano-fertilized, unfertilized, and wild plants remain unknown. Further, an archaeological reference for regional marine $\delta^{34}$S values is lacking [22].

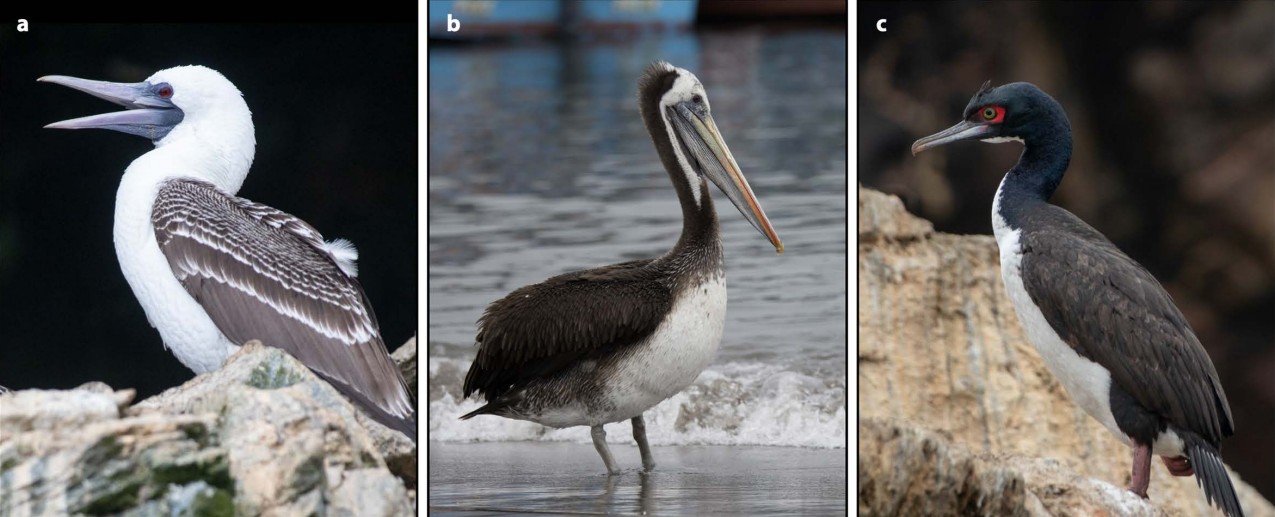

**Fig 2. The primary guano-producing bird species. (A)** *Sula variegata* (Peruvian booby). **(B)** *Pelecanus thagus* (Peruvian pelican). **(C)** *Leucocarbo bougainvilliorum* (Guanay cormorant). Photos by Diego H. **(A** and **C)** and Claude Kolwelter **(B)**, iNaturalist.org. Licensed under CC-BY 4.0. Cropped from originals.

We conduct stable carbon, nitrogen, and sulfur isotopic analyses of 35 maize cobs and 11 seabirds from cemeteries and a fishing settlement spanning the late Formative period (*c*. 200 BCE – 150 CE) to the Colonial period (1532–1825 CE). By integrating isotopic, archaeological, and historical data from Chincha, we provide some of the strongest evidence to date for pre-Inca seabird guano fertilization in the Peruvian Andes, thereby demonstrating the critical role of seabirds to the sociopolitical and economic expansion of the Chincha Kingdom. More broadly, our findings nuance understandings of agricultural sustainability in coastal environments and highlight marine fertilizers as potentially significant drivers of social change among pre-Hispanic societies throughout the Andes.

## Results

Recent archaeological research in southern Peru revealed a permanent shoreline fishing settlement at Jahuay [23,24] and a dense array of mortuary sites in the middle Chincha Valley [25–27]. Originally occupied during the late Formative period [23,24], Jahuay was reoccupied as a cemetery with middens from the LIP to the Colonial period. Deposits include diverse marine faunal remains, including seabirds, and a pyro-engraved gourd with seabird iconography (Fig 3). Regional survey identified over 500 graves that cluster into 44 cemeteries [25,26]. We documented maize cobs in over 80 graves from at least 15 cemeteries [27,28], suggesting maize nourished both the living and the dead (S1 Fig in S1 File).

### Written sources

Written sources describe the acquisition of seabird guano and the importance of this resource for trade and food production during the pre-Hispanic and colonial eras. Early colonial chronicles report that in coastal valleys from at least northern Peru down to northern Chile, groups sailed to nearshore islands on rafts and brought back abundant bird droppings for the cultivation of crops, including maize [29]. For example, the Lunahuaná people, located in the Cañete Valley north of Chincha, manured their agricultural fields with guano [30]. The Lunahuaná ethnonym may be derived from "guano" and/ or "guanay" [31]. Guano was incorporated into soils at the time of planting or shortly afterward and was reportedly applied twice [32]. Swiss naturalist Johann Jakob von Tschudi observed the fertilizer being used alongside irrigation, noting that

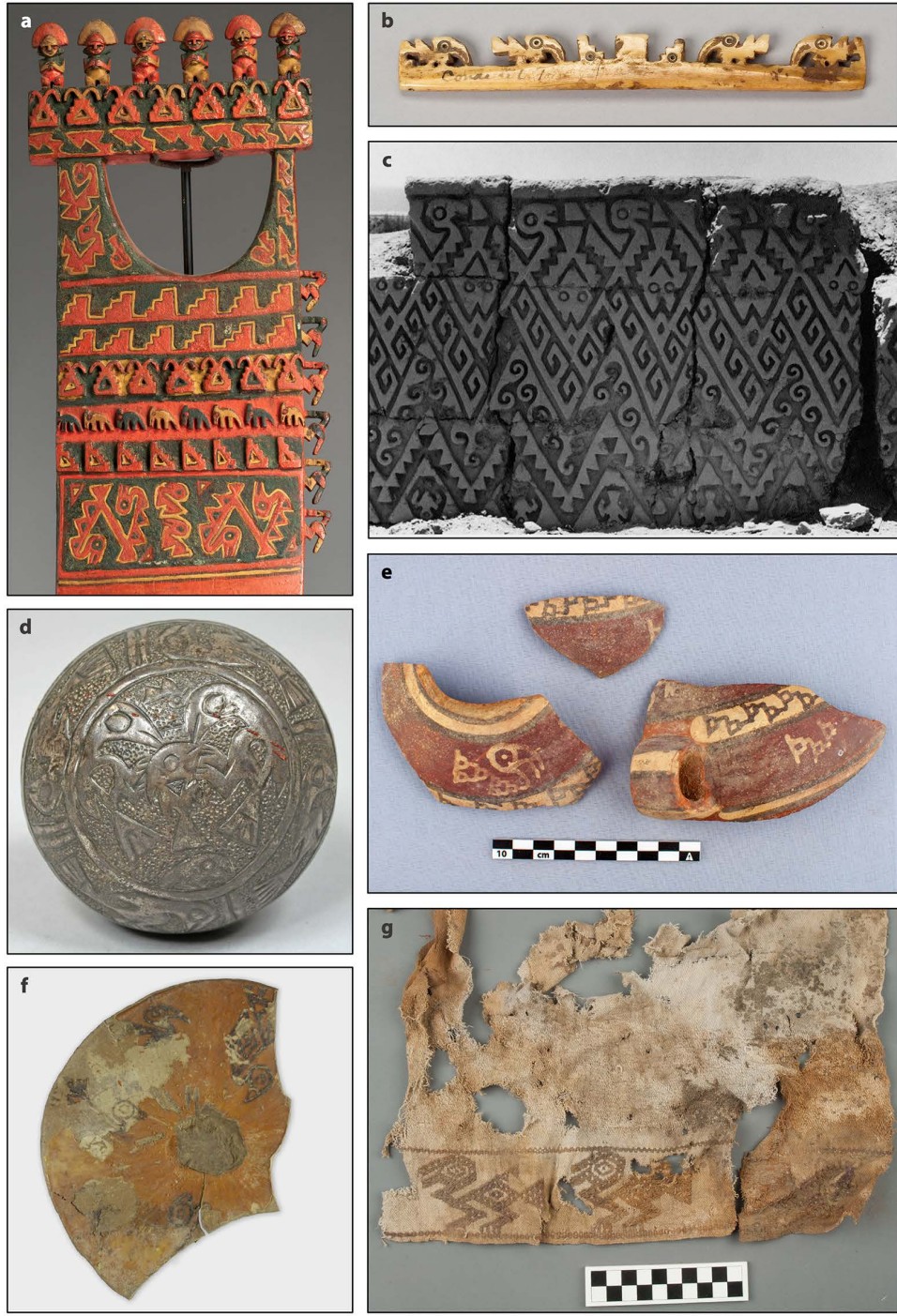

**Fig 3. LIP and LH seabird imagery from the Peruvian southern coast. (A)** Ceremonial digging stick or paddle, The Met Museum 1979.206.1025. **(B)** Bone balance-beam scale, The Art Institute of Chicago 1955.2579d. **(C)** Adobe frieze (now destroyed) at the site of La Centinela c.1938, Bennett Greig (1907–1944). **(D)** Embossed lead and silver ball depicting seabirds eating a fish, The Met Museum 82.1.22. **(E)** Ceramic jar from UC-018 mortuary site, middle Chincha Valley, photo by J. Bongers. **(F)** Pyro-engraved gourd from Jahuay, Quebrada de Topará, photo by J. Osborn. **(G)** Embroidered textile from UC-25, middle Chincha Valley, photo by C. O'Shea. (A-D): CC0 Public Domain. All photos cropped from the originals.

a fist-sized amount was added to each plant, and entire fields were subsequently submerged in water [33]. Guano could restore barren agricultural lands and boost maize production, making it an essential resource for sustainability [29]. Consequently, it became a highly sought-after item that communities traded among each other [29]. The Inca highly valued this fertilizer, imposing access restrictions on the guano islands during the breeding season and forbidding the killing of guano birds, on or off the islands, under penalty of death [34,35].

## Material culture

LIP and LH iconography from Chincha and its neighboring valleys highlights the prominent role of marine avifauna and demonstrates that Chincha residents had established knowledge of the interconnection between seabirds and maize fertility (Fig 3). Seabirds feature in textiles [25,36], ceramics [37,38], balance-beam scales [39], spindles [37], decorated gourds [23], gold and silver metallurgy [37], adobe friezes and wall paintings [40,41], and ceremonial wooden boards, possibly used for digging or paddling watercraft [42,43]. Some naturalistic depictions leave little doubt as to their identification. More abstract forms can still be confidently identified as seabirds due to their curved necks, long beaks, and their frequent association with fish and wave motifs. Geometric rhomboid forms may also represent abstracted birds and fish [37] (S2 Fig in S1 File). Possible maize is shown sprouting from abstracted fish and stepped-terrace motifs (Fig 3A and S2 Fig in S1 File), linking marine fertility with agricultural productivity. Seabird imagery identified in Chincha middle valley cemeteries [25] (Fig 3E and Fig 3G) and at agricultural centers [39] reflect the far-reaching symbolic and economic importance of marine resources.

## Faunal remains

A high frequency of guano birds in LIP and LH zooarchaeological assemblages from the Peruvian southern coast suggests local groups regularly consumed these animals. In a review of LIP and LH zooarchaeological assemblages from the south coast (S1 File), cormorants (Phalacrocoracidae) were consistently the most abundant taxon identified, comprising 56.73% of the total NISP. Gannets and boobies (Sulidae; 14.32%) and pelicans (Pelecanidae; 14.21%) were ranked second and third respectively.

## Faunal isotopic data

Bone collagen from 11 guano birds (5 guanay cormorants, 3 Peruvian boobies, 1 Peruvian pelican, 1 gull, and 1 penguin) from late Formative and LH contexts at Jahuay [16,21] provide a regional isotopic baseline for Chincha seabirds (for quality control see S2 Dataset). $\delta^{15}$N values averaged +16.1±1.9‰ ranging from +14.7‰ to +21.4‰, $\delta^{13}$C values average −11.6±0.09‰ and range from −9.9‰ to −13.0‰, and $\delta^{34}$S values average +16.9±0.55‰ with a range of +15.0‰ to +17.3‰. After correcting for a generalized $\Delta_{diet-tissue}$ offset of ~+3.4±1‰ for $\delta^{15}$N [44], we suggest the $\delta^{15}$N data are consistent with modern fishmeal values ($\delta^{15}$N ~+13‰) reported from this latitude (Pisco) [21].

## Radiocarbon data

Twenty $^{14}$C AMS (accelerator mass spectrometry) dates from eleven graves sampled for maize overlap with the LIP, LH, and Colonial period (S1 Dataset). We dated tooth dentine, bone collagen, reeds, and hair from these contexts. After correcting for the marine reservoir effect (see Methods), these graves date from c. 1155 to 1675 CE (at 95.4% confidence). Median calibrations fall between c. 1195 to 1570 CE.

## Maize carbon and nitrogen isotopes

As a pillar of present-day global agriculture, maize is well-studied isotopically. Despite unburned maize cobs sometimes producing unreliable radiocarbon ages [45], charred and desiccated archaeological remains appear to retain reliable

stable isotopic values [15,46]. Further information on maize isotope physiology can be found in Supplementary section S2 Dataset.

We analyzed 35 surface-collected maize cobs collected from 26 tombs across 14 Middle Valley cemeteries for $\delta^{13}$C and $\delta^{15}$N. All cobs were more than 50% complete. One hundred percent of the sample was run in duplicates or triplicates. All isotopic data for individual and averaged replicates are reported in S2 Dataset. The %C, %N, and atomic C:N ratios are within the range of previously reported archaeological maize data (S3 Fig in S1 File) [10,46]. Ranges of averaged replicates for maize ($n = 35$), are −11.7‰ to −8.7‰ for $\delta^{13}$C (mean = −10.0‰ ± 0.6) and +10.0‰ to +27.4‰ for $\delta^{15}$N (mean = +19.4‰ ± 4.1). The Shapiro-Wilk test of normality indicates that $\delta^{13}$C ($W(25) = 0.893$, $p = 0.013$) deviates from a normal distribution while $\delta^{15}$N ($W(25) = 0.9617$, $p = 0.429$) does not.

All archaeological maize has $\delta^{15}$N values consistent with growing conditions altered by some type of soil amendment (dung, char, guano, etc.) [18]. At least thirteen maize samples have $\delta^{15}$N values equal to or higher than +20‰, a conservative lower limit for identifying guano based on experimental studies [18] and the local nitrogen expectations established by our archaeological guano bird values. Nine additional samples have $\delta^{15}$N values between +15‰ and +19.9‰, suggesting a majority ($n = 22$) of the sample could represent crops that were amended with fertilizers. Maize samples with $\delta^{15}$N values at or above +20‰ include cobs directly associated with dates from the LIP ($n = 2$).

## Maize sulfur isotopes

Twenty of the 35 maize cobs were analyzed for stable sulfur at 100% duplication. The mean SD for reference material replicates was ± 0.46‰ and the mean absolute accuracy for calibrated reference materials was ± 0.17‰. After averaging replicates, $\delta^{34}$S ranged from +1.9‰ to +6.2‰ (mean = +3.9‰ ± 1.1), and mean Total S was 8.3 µg. With one exception, MCV-296, the precision for the replicate samples was better than 0.5‰ ($\delta^{34}$S) and 0.01%S, which fell within the typical range of precision for sulfur [47]. The %S for samples ranged between 0.02% to 0.28% with an average of 0.09% ± 0.06. Published %S for modern maize kernels ranges from 0.09–0.13% [16]. No good quality control parameters exist to assess $\delta^{34}$S values from archaeological maize. As such, we interpret the sulfur data with caution. Nitsch et al. [48] have suggested that sulfur in plants is likely affected by diagenetic processes similar to nitrogen, as both elements are measured from the amino acids comprising protein. They observed that while the total concentrations of sulfur (%S) may decrease due to various diagenetic processes, there were no significant changes in $\delta^{34}$S values [47]. In this study, three archaeological maize samples (0.02–0.04%) fell below the minimum known sulfur concentration for modern plants (>0.05%) [49]. While samples with a lower %S do not necessarily correlate to affected $\delta^{34}$S values [48], lower elemental concentrations can prove difficult to measure accurately; therefore, we interpret only values with ≥0.05 %S. $\delta^{34}$S values of the samples meeting these criteria ranged from +2.6‰ to +6.2‰ with a mean of +3.8‰ ± 0.9. These $\delta^{34}$S values are consistent with ranges measured for guano-fertilized maize reported in a previous experimental study [16]; however, none of these values reach the range of $\delta^{34}$S values documented for maize growth chamber studies [16], our archaeological seabirds (+15.0‰ to 17.3‰) or crops amended with other known marine fertilizers [17].

## Comparisons among archaeological and modern maize

Chincha maize $\delta^{13}$C and $\delta^{15}$N values are consistent with measurements of archaeological maize from another dataset [10] henceforth identified as "Chile", which identifies fertilization with seabird guano from the LIP period onwards (Fig 4). While radiocarbon date ranges may span multiple periods (S1 Dataset), time periods assigned to Chincha maize cobs are based on modeled median calibrations from sampled tombs and the means of modeled median calibrations when these tombs have multiple dates. We restrict our comparison to maize cobs from Chile to limit any effects of differential uptake by different plant parts [8]. Shapiro-Wilk tests of normality showed that neither $\delta^{13}$C ($W(101) = 0.903$, $p < 0.001$) nor $\delta^{15}$N ($W(101) = 0.949$, $p = 0.001$) of the Chile maize cobs were normally distributed, therefore we employ non-parametric

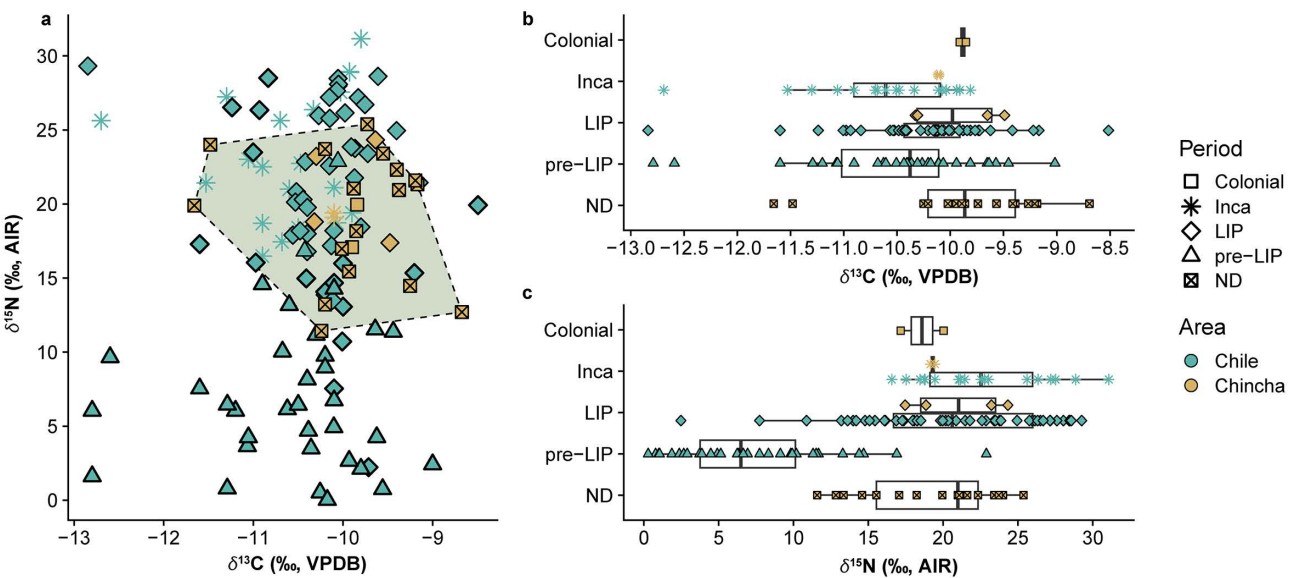

**Fig 4. Comparison of δ¹³C and δ¹⁵N of archaeological maize from Chincha and Chile [10].** Scatterplot of δ¹³C and δ¹⁵N with a convex hull constructed for the range of Chincha samples **(A)** and box-and-whisker plots of δ¹³C **(B)** and δ¹⁵N values **(C)** organized by period (ND = no date/maize from undated context).

Mann-Whitney *U* tests in our comparisons. These analyses show archaeological Chincha (mean = +19.6‰ ± 4.0, *n* = 25) and Chile (mean = +16.6 ± 8.9‰, *n* = 101) maize have similar δ¹⁵N values (*U* = 1058.5; *p* = 0.215). This remains true (*U* = 177.5; *p* = 0.646) when both Chincha (mean = +20.5 ± 2.7‰, *n* = 6) and Chile (mean = +21.3 ± 5.6‰, *n* = 67) samples are restricted to maize cobs dated to the LIP and Inca period. δ¹³C values of Chincha (mean = −10.0 ± 0.4‰, *n* = 6) and Chile (mean = −10.3 ± 0.7‰, *n* = 67) from LIP and LH contexts are almost identical (*U* = 135.5; *p* = 0.190) as well. Chincha maize δ¹⁵N values are significantly higher (*U* = 34; *p* < 0.001) than Chile maize cobs from the pre-LIP (mean = +7.3 ± 5.2‰, *n* = 34). Separately, LIP-dated Chincha (mean = +21.0 ± 3.3‰, *n* = 4) and Chile (mean = +20.6 ± 6.0‰, *n* = 48) maize have very similar δ¹⁵N values, while in Inca period contexts, Chile maize (mean = +22.9 ± 4.2‰, *n* = 19) has somewhat higher δ¹⁵N than the Chincha sample (mean = +19.4 ± 0.2‰, *n* = 2). δ¹³C of maize cobs from Chincha and Chile from both the LIP and Inca periods are consistent (S2 Table in S1 File). Low sample sizes from Chincha prevent formal statistical comparison of individual time periods.

No regional studies have used δ³⁴S to examine archaeological fertilization practices; therefore, Chincha maize δ³⁴S values were compared to modern experimental values [16] and are interpreted with caution. The stable isotopic compositions of the Chincha maize samples are most similar in δ¹⁵N and δ³⁴S bivariate isotope space to maize plants fertilized with seabird guano in controlled field experiments [16]. The comparative experimental sample only contains data for leaves and grains. While different parts of maize plants may yield different isotopic values due to variation in uptake residence and pathways, there is some research to suggest little difference in δ¹⁵N values (<0.5‰) between cobs and grains based on unfertilized and sheep-manured maize kernels and cobs [50].

Among experimentally grown maize, plants amended with seabird guano occupy the greatest amount of isotope space, while those fertilized with camelid dung or a modern ammonium sulfate mixture cover a much narrower region (S3 Table in S1 File). The isotope space occupied by guano fertilized plants encompasses an almost entirely different region of bivariate space than the other experimental fertilizer conditions (Fig 5). Both standard ellipse area (SEA) and kernel utilization density (KUD) overlap are generally in agreement with the SEA model indicating isotope space overlap of 2% or less and the KUD model showing no overlap of camelid dung and/or ammonium sulfate fertilized plants with

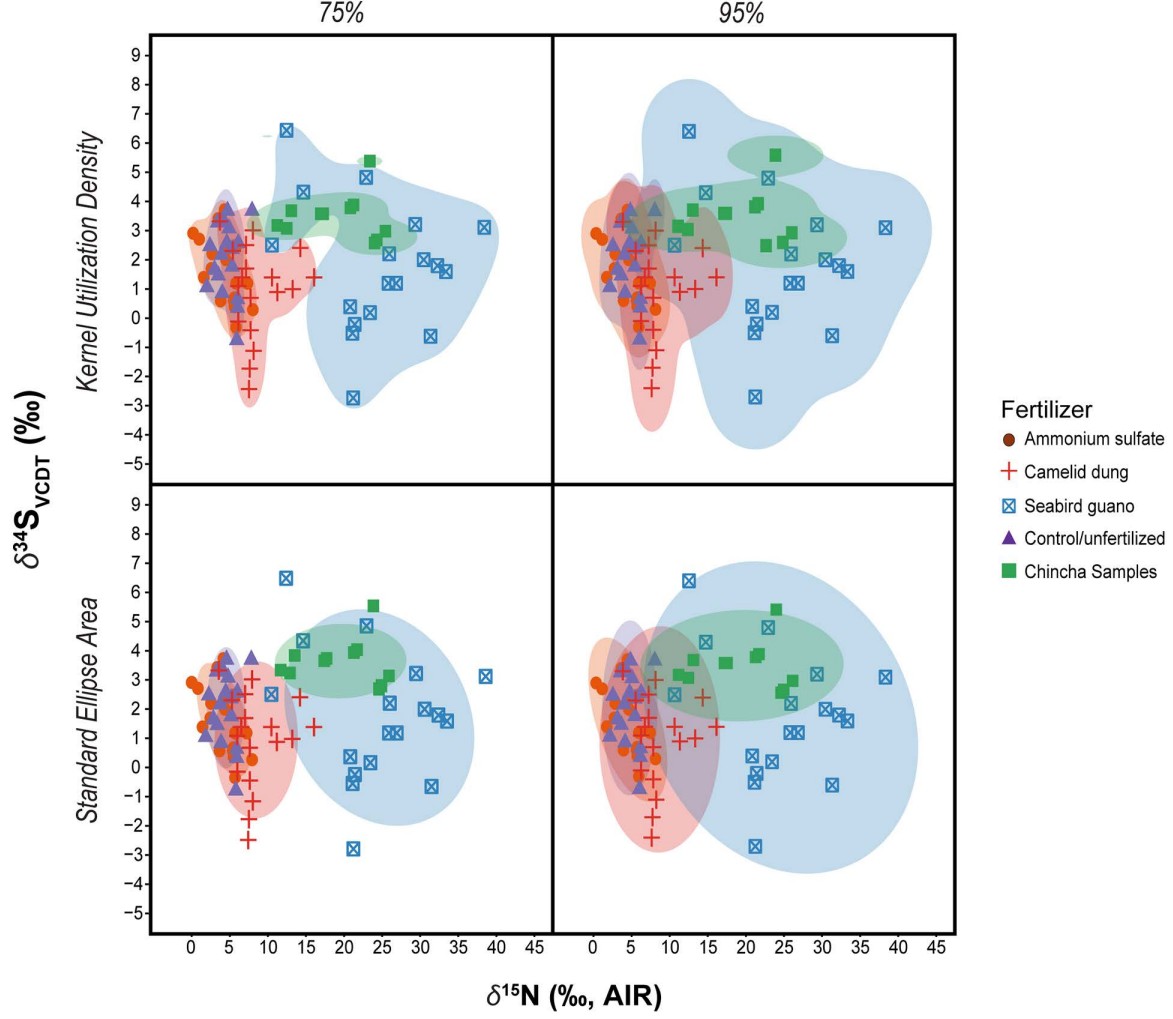

**Fig 5. Isotope space size and overlaps.** Plotted isotope space kernel utilization density functions (KUD; top row) and standard ellipse areas (SEA; bottom row) comparing Chincha maize with maize grown under different experimental fertilizer conditions [16]. Columns display results at commonly selected contour levels – 75% and 95%.

seabird guano fertilized maize at the 75% contour level. At the 95% contour level maize plants fertilized with camelid dung overlap a small range (~ 15%) of the isotope space covered by plants fertilized with seabird guano (S4 Table in S1 File). The ammonium sulfate, camelid dung, and unfertilized conditions result in plants with largely overlapping isotope space, although camelid dung fertilized plants show some expansion on the $\delta^{15}$N axis (Fig 5). There is very little overlap (≤ 12% at the 95% contour level) of the Chincha maize samples by the experimental plants grown in unfertilized or ammonium sulfate fertilizer conditions. There is virtually no overlap (3%) in the isotope space of Chincha maize by plants fertilized with camelid dung at the 75% contour level. This overlap increases to ~20–25% (depending on model) at the 95% contour level, well below the cut-off of 60% that would indicate a strong likelihood of occupying the same isotope space. On the other hand, the isotope space occupied by experimentally fertilized seabird guano plants covers ~85–90% of the area covered by the Chincha maize at the 75% contour level and entirely overlaps (100%) the Chincha sample isotope space at the 95% level (S4 Table in S1 File).

## Discussion

This study makes methodological and empirical advances to the nascent study of agricultural fertilization in the Andes. Results suggest Indigenous communities used marine fertilizers for maize cultivation by at least 1250 CE. We demonstrate that stable sulfur analysis remains crucial for differentiating between past marine and terrestrial diets. Our multidisciplinary dataset provides strong support for pre-Inca seabird guano fertilization, an effective agricultural practice for boosting crop production that is more commonly associated with industrial societies. This likely contributed to the rise of the Chincha Kingdom and enhanced its strategic importance for the Inca Empire.

### Shifting isotopic expectations for guano fertilization

Local isotopic compositions of seabird guano vary due to basal marine ecology, variation in bird diets, and chemical alterations over time (diagenesis) [21], therefore, regional bird faunal data may help estimate the upper and lower limits of guano inputs for fertilized plants. Fishmeal measured from different latitudes along coastal South America suggests that $\delta^{15}N$ values in baseline fish ecology increase significantly with latitude (+6.5‰ over 9 degrees of latitude), suggesting guano birds at higher latitudes may have diets more enriched in $^{15}N$, which would in turn lead to higher $\delta^{15}N$ values of their middens and guano. Additionally, the chemical composition of guano can change within years of initial excretion. Generally, guano %N decreases over time while becoming more enriched in $^{15}N$ (up to +30‰ for fossil guano) [21]. Our archaeological faunal data provide an important approximation of the possible lower end of guano isotopic values in the Chincha area and, additionally, a control for possible industrial changes to marine sulfur reservoirs [47]. Our data suggest that input fertilizer $\delta^{15}N$ values above ~+20‰ are likely reasonable for Chincha, especially if fresh guano was used [21]. Further, local guano variation—or aridity—may explain the slightly higher range of $\delta^{15}N$ values observed in Chile, where the baseline $\delta^{15}N$ soils and guano should be a few per mil higher [10]. Our measured $\delta^{34}S$ range for seabirds (+15.0 to 17.0‰) sets a lower expected end range for marine sulfur than previously suggested (+20‰) [16,47,51].

### Implications for regional paleodietary reconstructions

Reliably identifying marine isotopic "contamination" in terrestrial food systems is essential for improving paleodietary reconstructions and radiocarbon calibrations in the Central Andes region [52]. Szpak et al. [8,11] have noted that in archaeological periods where agriculture contributed significantly to human and animal diets, the $^{15}N$ enrichment caused by marine fertilizers could introduce equifinal pathways to paleodietary interpretations of $\delta^{13}C$ and $\delta^{15}N$. Specifically, terrestrial $C_4$ resources enriched in the heavier nitrogen isotope can mimic the 2D isotope space of marine dietary sources [8], as shown in Fig 6A.

However, it is critical to caveat that the isotopic values measured from bone collagen and hair keratin primarily reflect dietary protein but, generally, the bioavailable protein in $C_4$ plants, such as maize, is low, unless chemically altered through cultural practices like nixtamalization [54]. Therefore, terrestrial $C_4$ crops enriched in $^{15}N$ from marine products would only significantly alter human collagen values if they entered the food chain via animals who consumed fertilized crops. Elsewhere in the Andes stable sulfur ($\delta^{34}S$) has been applied to discriminate between terrestrial and marine dietary inputs in contexts where $\delta^{13}C$ and $\delta^{15}N$ could not provide clear distinctions [51,55]. While experimental studies have hypothesized that plants fertilized with marine products should be enriched in $^{34}S$ relative to unfertilized plants, experimental maize $\delta^{34}S$ data, and the archaeological $\delta^{34}S$ measurements presented here, do not consistently support this expectation [16,17]. At present it remains unclear whether marine-fertilized terrestrial plants could become so enriched in $^{34}S$ that they would enter marine isotope space [16].

Without reliable quality control parameters for evaluating $\delta^{34}S$ measurements from archaeological plants, we have elected to limit interpretations of our data. However, given review of agricultural literature, we suggest that our low $\delta^{34}S$ values may be realistic, due to differences in plant uptake of nitrogen and sulfur, which merit further study. For example, sulfur uptake is influenced by parent soil composition, local climate and atmospheric contributions of $\delta^{34}S$, watering

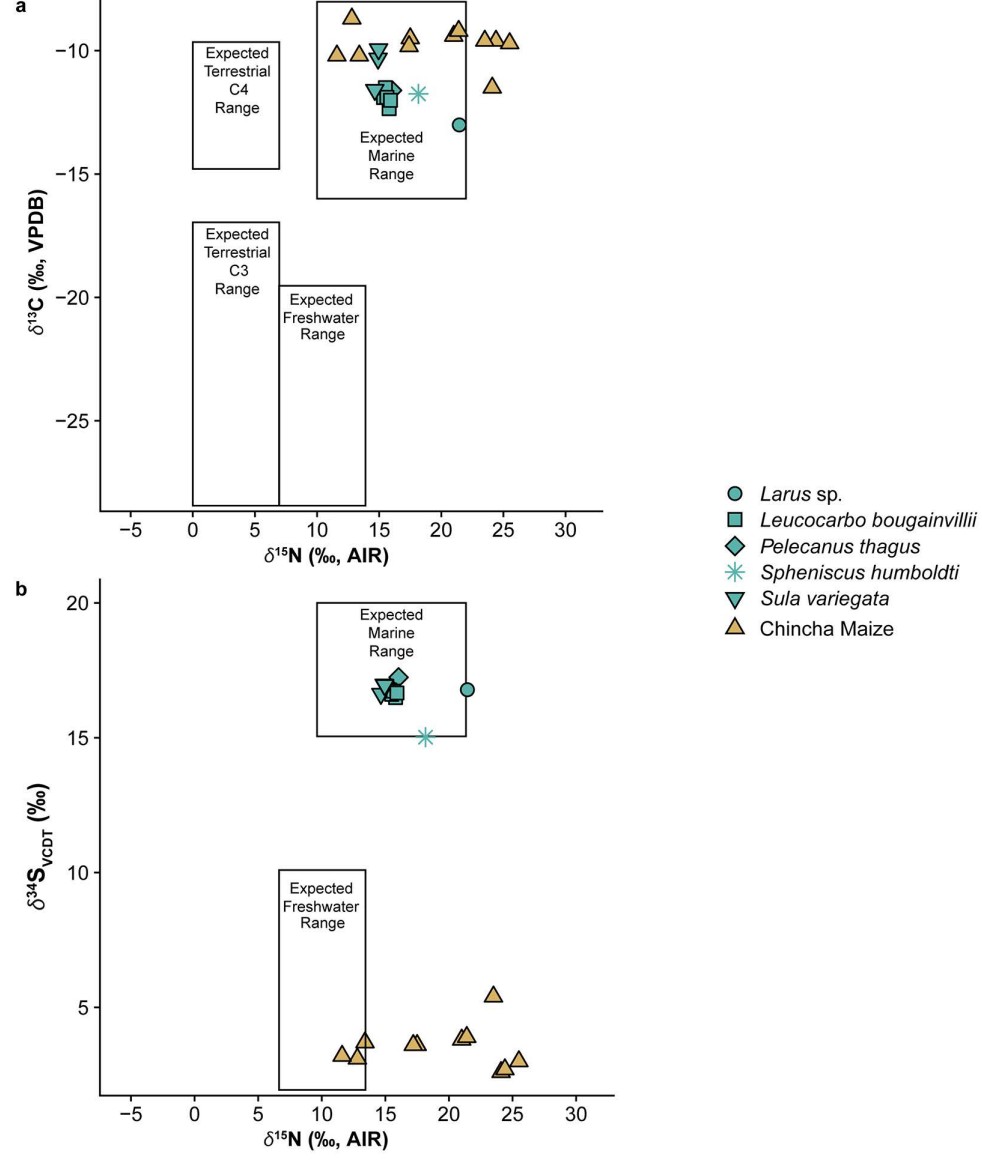

**Fig 6. Basic schematic of how marine "contaminated" plants can create equifinality issues for paleodietary interpretations.** The plot compares Chincha maize (C$_4$ crops) artificially enriched in [15]N and with the isotopic measurements from archaeological guano birds (this study), demonstrating the potential utility of $\delta^{34}$S for discriminating between terrestrial plants with marine products and marine resources. **(A)** $\delta^{13}$C and $\delta^{15}$N data with boxes for expected terrestrial C$_4$ and C$_3$, freshwater, and marine ranges [53]. **(B)** $\delta^{34}$S and $\delta^{15}$N data with boxes for expected freshwater and marine ranges indicated [47].

practices, and the quantity and timing of fertilization [49]. It is notable that sandy soils, like those in the Chincha Valley, are susceptible to sulfur deficiencies and leaching [49], which is exacerbated in irrigated fields where heavy inundation with water may cause translocation of sulfur [49]. Our historical records highlight cultural practices of flooding fields after fertilization, which may provide a cultural explanation for reduced guano-derived $\delta^{34}$S uptake in the valley [8]. It is further worth noting that highly negative $\delta^{34}$S values have been measured from sediments along the west coast of the South American continent and these values may compete with $\delta^{34}$S values from marine amendments [56].

If high $\delta^{34}S$ values (>+15‰) are not reliably routed from marine fertilizers to land crops, $\delta^{34}S$ measurements from human and animal tissues could still provide a critical method to address paleodietary equifinality unknowns introduced by marine fertilizers, as we show with Fig 6B [51,55]. Therefore, resolving sulfur uptake in crops and developing quality control parameters for archaeological plants are important and viable areas of future research.

## Seabird guano fertilization: an Indigenous land management practice in the Andes

Archaeological research suggests guano fertilization began as early as the 1st millennium AD on the Peruvian north coast [11,57], but this remains to be tested isotopically. Written sources record the persistence of guano soil amendments through the Colonial period [32]. Our multidisciplinary data carry significant implications for understanding the regional significance of maize agriculture and seabird guano and demonstrate how local knowledge of land management strategies were potentially represented and reinforced through iconography. Successful maize crop cultivation necessitates specific agricultural strategies such as soil amendments (e.g., animal fertilizers, biochar, and composts), companion planting (with nitrogen-fixing plants, such as beans), burning or tilling of fields, or crop rotation [2,18]. It is likely, therefore, that Indigenous land management practices, such as manure fertilization, have long been important to regional maize cultivation, as has been documented for other ecosystems and natural resources in the Americas [58]. Documented maize fertilization practices vary, with methods such as companion planting, field burning, and guano application overlapping with major domestication centers [1,59]. In Peru, historical and archaeological data suggest guano was preferentially applied to maize. Because sulfur is a limiting nutrient that can impede maize growth [49], it is possible preferential application of guano reflected regional knowledge of differential nutrient requirements among plant species, supporting regional domestication processes.

The Peruvian coastal desert, encompassing numerous riverine valleys fed by Andean runoff, enabled irrigation agriculture that became an economic cornerstone for large polities including not only Chincha, but also Moche and Chimú. Archaeological findings from the northern guano islands suggest that the earliest group to interact with the islands were the Moche during the 1st millennium AD [57]. Given the adverse effects of irrigation, aridity, and field reuse on soils, and the accessibility of the Peruvian guano islands, it is plausible that guano fertilization shaped the economic and sociopolitical expansion of multiple coastal Andean societies. This hypothesis can be tested using our multidisciplinary research design, which is applicable across the Andes and especially in coastal valleys such as Virú and Lurín, which are located near guano-rich islands.

The integration of bird imagery throughout the Chincha iconographic corpus emphasizes that traditional ecological knowledge was interwoven into local cultural and religious expression. Prominent bird iconography is associated with potential maize imagery on agricultural digging tools and in administrative spaces. Seabirds were recognized as the producers of a valuable resource and respected accordingly. Chincha specialists would have understood that guano birds are susceptible to mass die-offs from El Niño-related disruptions to the marine food chain [60]. With time, this ecological knowledge may have shaped the conservation efforts formalized under Inca rule [34,35].

## Seabird guano contributed to the development of the Chincha Kingdom

These results enhance understandings of pre-Hispanic agricultural sustainability in the Peruvian coastal desert, one of the driest regions on earth, and how privileged access to guano may have shaped the development of the Chincha Kingdom and its relationship to the Inca Empire. Sustainable agricultural practices involving maize production can form the economic foundation of large-scale societies by ensuring reliable food resources that support population growth [61–63]. Seabird guano fertilization in Chincha appears to have been no exception. This land management strategy would have been essential to increasing maize crop yields and supporting the dense LIP and LH settlement patterns in the Chincha Valley [13], as well as a burgeoning population of 30,000 tribute payers [12] and likely over 100,000 people total. In this sense, the use of seabird guano fertilization to sustain growing coastal populations provides ancillary support for the Maritime

Foundations of Andean Civilization hypothesis [64]. Our work demonstrates that by at least the 13[th] century, local farmers in the Chincha Valley manured irrigated fields with guano for maize cultivation. Precisely when and where Andean peoples first identified the agricultural potential of guano fertilization remains open to future investigation. Such a practice would have optimized the production of maize by mitigating NPK and sulfur depletion in soils caused by repeated use, aridity, and irrigation.

Guano likely constituted a critical source of economic and political power that expanded trade networks and facilitated cooperation among Chincha's specialized farmers, fisherfolk, and merchants. Previous models [12,65] argue that Chincha merchants in the LIP undertook maritime voyages to exchange for spondylus shells (*Spondylus princeps*), a highly valuable mollusk employed in rituals across the Andes [66], but limited LIP evidence of spondylus suggests it was not initially a source of wealth [67]. Instead, this research provides support for Marco Curatola's [14] model that guano was the primary driver of economic prosperity and sociopolitical influence for Chincha, and an important nexus for its network of interdependent specialists. In this view, fisherfolk acquired guano and provided it to farmers for maize production and to merchants for trading along the coast and into the highlands, expanding Chincha's agricultural productivity and mercantile influence. We cannot rule out that increased maize crop yields and population levels may have also intensified competition among groups or contributed to new forms of territorial control, but these hypotheses need to be further explored archaeologically. Results provide indirect support for Chincha seafaring capacity because acquiring guano from offshore islands necessitated ocean travel [14,32]. Balsa rafts were likely used to transport the fertilizer from the Chincha islands to the mainland. Colonial-era sources report the Chincha lord commanded 100,000 rafts [68]. Chincha's maritime knowledge and access to the Chincha islands underscore the polity's strategic importance for the Inca Empire.

The Inca incorporated Chincha into their empire after a "peaceful" capitulation [29], creating one of the few calculated alliances of its kind for this time. Some argue that the agreement resulted in Chincha gaining access to the spondylus trade in exchange for voluntary subjugation and tribute payment [67]. We propose that guano may have played a key role in these negotiations. Land management strategies that boosted the productivity of crops, especially maize, were central for Inca expansion [61]. The Inca valued maize as a staple crop and an essential ingredient for fermented beer (*chicha*) that was consumed during important ceremonies [69]. Employing Chincha as a seafaring client state to gain control over guano would have advanced state production of maize in coastal and highland environments. Inca roads and administrative centers found in Chincha and the nearby Pisco Valley could have facilitated the movement of guano throughout the empire. The Inca demand for guano [35] strengthened Chincha's negotiating power, allowing them to secure a prominent position within Inca society. Indeed, the strength of Chincha-Inca relations is best illustrated by observations made before Francisco Pizarro's fateful attack on Andean people at Cajamarca in 1532 CE: the Chincha lord was the only other person being transported on a litter near Atahualpa himself, one of the final rulers of the Inca Empire [68].

## Conclusions

Our multidisciplinary approach contributes some of the strongest evidence yet for pre-Inca seabird guano fertilization in the Peruvian Andes, with important implications for understanding sociopolitical and economic transformations throughout the broader region. Contextualized with recent datasets from Peru and Chile, this research suggests that guano fertilization was a widespread and enduring Indigenous land-use practice that contributed to the rise of the Chincha Kingdom and potentially other large pre-Hispanic societies along the coast. As demonstrated here and elsewhere, stable isotopic analysis of $\delta^{15}N$ can provide a reliable method of detecting past guano fertilization and is essential for cases where material culture is not available. Wider geographic coverage of archaeological crop nitrogen could potentially document variations in guano $\delta^{15}N$ associated with latitude, aridity, and age of the deposit. While not always available or preserved, we found regional faunal isotopic data, written sources, and iconography to be essential for nuancing our isotopic interpretations and for understanding the importance of marine ecosystems to the expansion of the Chincha Kingdom. This multidisciplinary approach demonstrates the crucial need to integrate multi-isotopic data with other lines of evidence because of the

lack of quality control criteria and unresolved questions around differential plant uptake pathways for $\delta^{13}C$, $\delta^{15}N$, and $\delta^{34}S$ of archaeological plant remains. Looking ahead, widespread analyses spanning multiple periods will be critical for identifying the origins and expansion of guano fertilization and how it impacted the long-term development of societies throughout the Andes.

## Materials and methods

### Stable isotope analysis of maize

Approximately 3-millimeter samples of desiccated and uncharred maize cob fragments were prepared at the Skeletal and Environmental Isotope Laboratory (SEIL) at the University of California, Merced following established protocols for the stable isotope analysis of archaeobotanical maize [46]. The necessity and protocols for pre-treating desiccated and carbonized plants is debated [70]. We elected not to pre-treat for potential humic contamination, as the effect on $\delta^{13}C$ values is minimal (<1‰) and the recommended HCl rinse at 80°C for 30 minutes can result in significant sample loss [70,71]. No FTIR was used to test for contaminants, as this practice can be imprecise and unnecessarily destructive; rather, as this study is focused on fertilization, which requires reliable $\delta^{15}N$ values, all samples were pre-treated for possible nitrate contamination following [71]. Cobs were sonicated in Millipore water (changing every 20 minutes) until clean, dried overnight at 30°C, and then homogenized into powder in methanol-cleaned canisters on the bead mill.

We prepared duplicate and triplicate samples, where sample availability allowed, of ~1.0 mg (1.1 ± 0.08 mg averaged across 73 replicates) of powdered maize for stable carbon and nitrogen analysis in tin capsules. Due to low sample availability after pre-treatment, a subset of nine maize cobs were prepared in duplicate and triplicate at a target weight of 0.4 mg (0.41 ± 0.02 mg averaged across 27 replicates), which is consistent with masses analyzed in other regional studies [46]. While the $\delta^{13}C$ (mean = −10.4‰ ± 0.3) and $\delta^{15}N$ (mean = + 19.2‰ ± 5.3) values of these samples are comparable with the other 26 maize cob samples analyzed at a higher mass of 1.1 ± 0.08 mg ($\delta^{13}C$ mean = −9.9‰ ± 0.9; $\delta^{15}N$ mean = + 19.5‰ ± 3.8), we exercise caution by restricting our interpretations to only samples with a target mass of 1.0 mg due to recent concerns regarding the reliability of samples with low masses [70]. One maize sample, MCV-262, was excluded from analysis because its context and date are unknown. Samples for stable sulfur analysis were prepared as 9–10 mg duplicates, measured into tin capsules.

Staff at the Stable Isotope Ecosystem Laboratory (SIELO) at the University of California, Merced measured $\delta^{13}C$ and $\delta^{15}N$ and elemental carbon and nitrogen contents of maize on a *Costech 4010 Elemental Analyzer* coupled with a *Delta V Plus Continuous Flow Isotope Ratio Mass Spectrometer* alongside standard reference materials. SIELO determined elemental carbon and nitrogen contents through linear regression of $CO_2$ and $N_2$ sample gas peak areas against known carbon and nitrogen contents of in-house (peach leaf) and international standards (USGS 40, EA acetanilide, USGS 41a, and Costech acetanilide). SIELO corrected raw measurements for instrumental drift and mass linearity and standardized those values to the international VPDB ($\delta^{13}C$) and AIR ($\delta^{15}N$) scales using the USGS 41a and USGS 40 standard reference materials. All isotope compositions are expressed in standard delta notations where:

$$\delta = (R_{sample}/R_{standard} - 1) \times 1000$$

Staff at the Stable Isotope Facility (SIF), University of California, Davis, measured $\delta^{34}S$ and elemental sulfur in maize powder according to their 2023 protocols, using the Elementar vario ISOTOPE cube elemental analyzer interfaced to an Elementar PrecisION isotope ratio mass spectrometer. SIF combusted samples in a tungsten oxide-packed reactor, reduced gasses with elemental copper, buffered through quartz chips, and then separated $SO_2$ and $CO_2$ through adsorption columns for peak focusing before passing through to the IRMS for isotope ratio measurement. SIF applied post-run corrections for instrumental drift, corrected for oxygen variability using a regression of size references, normalized isotope ratios using bounded isotopic references, and calculated elemental totals based on IRMS peak area size references as

a calibration curve. SIF calibrated in-house references against international standards (IAEA-S-1, IAEA-S-2, IAEA-S-3, NBS-127, IAEA-SO-5, IAEA-SO-6), using cysteine, taurine, and salmon muscle for quality assurance and brightener, bovine gelatin, and blue-green algae for quality control. Isotope ratio measurements are reported as delta ($\delta$) relative to Vienna Canyon Diablo Troilite (V-CDT).

International standards, check standards, and replicate measurements are reported in S2 Dataset. Quality control and quality assurance statistics indicate better than 0.5‰ precision. At SIELO, the long-term (~5 years) reproducibility for in-house reference peach leaves is ~0.2‰ for both $\delta^{13}$C and $\delta^{15}$N, and long-term reproducibility across all sample types (soil, tissue, plants, etc.) is ~0.1‰ for $\delta^{13}$C and ~0.2‰ for $\delta^{15}$N. Only samples meeting in-house and broader quality metrics were included in analyses; one $\delta^{13}$C/$\delta^{15}$N data point was excluded due to a >1.0 SD within the triplicate measurement for $\delta^{15}$N.

Preliminary studies suggest that charred plants may have lower $\delta^{34}$S values (decreased by <1‰) and higher %S than uncharred crops. $\delta^{34}$S and %S appear to decrease over the life of the maize plant, and modern grains (late forming tissues) tend to show lower sulfur concentrations (between 0.10–0.13%) relative to other tissues (0.12–0.47%). Any %S values between 0.05–0.90% would fall within modern known ranges for plant sulfur concentrations [49]. There are presently no data to inform on $\delta^{34}$S$_{cob-grain}$ offsets, although, as later-forming tissues, we predict cob sulfur measurements should be close to grains.

## Stable isotope analysis of fauna

Archaeological bones were prepared at the University of Tübingen Biogeology Laboratory following an acid-base-acid extraction. Bone samples were prepared following [72]. Briefly, bones were cut into 300–500 milligram pieces, then cleaned using alternating a Millipore-acetone-Millipore soak under sonication at five-minute intervals. Samples were then rinsed with Millipore water under sonication until clean and dried at 35°C for 48 hours, then crushed and sieved through a <0.7-millimeter sieve. Samples were extracted following [72]. Between 250–450 milligrams of bone powder was transferred to 100 mL beakers and mixed in 40 mL of a 1M HCl solution for 20 minutes to dissolve the mineral structure before being filtered through Millipore filters and returned to the beaker with 40 mL of 0.125M NaOH solution at room temperature for 20 hours to remove humic contamination. The remaining material was then filtered, rinsed with a pH2 HCl solution, transferred to a glass tube with approximately 15–20 mL of pH2 solution, then left in an oven for 17 hours at 100°C. The sample was then filtered, with the liquid collected in pre-weighed and labeled glass vials. Vials were frozen for 24 hours, then freeze-dried. Aliquots of approximately 2.5 milligrams of collagen were weighed into 8.5 x 5 millimeter tin capsules for stable carbon, nitrogen, and sulfur analysis, and 0.4 milligrams of collagen were weighed into 5.5 x 3.5 millimeter tin capsules for carbon and nitrogen analysis.

Samples were measured with two in-house matrix-matched standards, camel and elk collagen, which were extracted in the same batch as the faunal samples, then calibrated relative to Standard Reference Materials (SRMs) USGS40 ($\delta^{15}$N = –4.52‰ and $\delta^{13}$C = –26.39‰), USGS41a ($\delta^{15}$N = +47.55‰ and $\delta^{13}$C = +36.55‰), IAEA-S-1 ($\delta^{34}$S = –0.30‰), IAEA-S-2 ($\delta^{34}$S = +22.62‰), and IAEA-S-3 ($\delta^{34}$S = –32.49‰). The Biogeology lab replicates 10% of all samples and analyzes two in-house standards for every ten samples. Samples were analyzed on an EA-IRMS at the University of Tübingen Geography facility. The combustion temperature was 1150°C and the reduction temperature 850°C, with a sample and TCD helium carrier gas flow at 230 ml per minute. Analytical error below 0.1‰, 0.2‰ and 0.4‰ (1σ) was determined for $\delta^{13}$C, $\delta^{15}$N and $\delta^{34}$S, respectively, across all analyses.

Stable isotope ratios of $^{13}$C/$^{12}$C, $^{15}$N/$^{14}$N and $^{34}$S/$^{32}$S are set relative to international standards (VPDB for carbon, AIR for nitrogen and VCDT for sulfur). The isotopic ratios are expressed using delta ($\delta$) notation as follows:

$$\delta^{13}C = \left( (^{13}C/^{12}C)_{sample}/(^{13}C/^{12}C)_{reference} - 1 \right) \times 1,000\ (‰)$$

$$\delta^{15}N \ = \ ((^{15}N/^{14}N)_{\text{sample}}/(^{15}N/^{14}N)_{\text{reference}} \ - \ 1) \ \times \ 1,000 \ (‰)$$

$$\delta^{34}S \ = \ ((^{34}S/^{32}S)_{\text{sample}}/(^{34}S/^{32}S)_{\text{reference}} \ - \ 1) \ \times \ 1,000 \ (‰)$$

Bird bone collagen met the three established quality control standards including appropriate: (1) atomic C:N (2.9–3.6), C:S (600±300), and N:S ratios (200±100), (2) %C (>20%) and %N (>10%), and %S values (~0.23%), and (3) collagen yield (>0.5–1.0%) [73]. We used the conservative upper limits for the atomic C:N ratio outlined for birds in Table 4 in [73,74].

## Radiocarbon dating

We report 20 $^{14}C$ AMS dates associated with sampled maize from middle valley mortuary sites (S1 Dataset). All radiocarbon dates were obtained from the Keck-CCAMS facility at the University of California Irvine (UCIAMS) using published methods [75]. The relationship between $\delta^{15}N$ compositions of human remains and marine diet carries critical implications for calibrating dates and thus remains a critical point of discussion in the archaeology of western South America. This is largely because of the marine reservoir effect (MRE), which can make dates derived from people who consumed nitrogen-enriched foods (e.g., marine organisms) appear older than they are [27]. Values of $\delta^{15}N$ provide a roughly linear scale of the relative importance of marine dietary resources, with ~+11.5‰ indicating a wholly terrestrial diet and ~+22.0‰ indicating a predominantly (~90 per cent) marine diet [27,28]. Here, we consider the $\delta^{15}N$ value of +15.0‰ as a baseline for a marine diet [76] in the Chincha Valley because of the arid conditions and local use of seabird guano fertilizer. Therefore, we calibrated eight dates from human hair, bone, and teeth using a mixture of SHCal20 and Marine20 [77] based on estimates ranging from 10 to 30 (± 10)% marine dietary component, depending on the $\delta^{15}N$ values. The ΔR value for the Paracas area (110±49) [78], the best available estimate for ΔR in Chincha, was recalculated to −32±58 according to the Marine20 curve. We calibrated reed dates according to the ShCal20 Southern Hemisphere calibration curve [79] using OxCal v4.4 [80].

## Statistical analyses

Isotope space overlap metrics (sometimes referred to as isotope niche space) are calculated for comparison of $\delta^{15}N$ and $\delta^{34}S$ from Chincha maize samples with experimentally fertilized modern maize [16]. Unlike traditional comparative statistics isotope space analyses offer the ability to consider isotopes of different elements at once in a single measure. This approach uses multi-dimensional spatial statistics to compare the total amount of space occupied by study groups – in this case archaeological maize samples from Chincha versus modern maize grown under known fertilizer regimes – and the degree of spatial overlap among groups [81–83]. The application of isotope space measures here is aimed at determining which (if any) practices of fertilization the archaeological maize samples from Chincha are consistent with, not for assessing any aspect of niche as the methods are more commonly used.

Isotope space measures are calculated with the 'rKIN' package [83] in R which offers the possibility of analyzing datasets using three models: minimum convex polygons (MCPs), standard ellipse area (SEA), and kernel utilization density (KUD). These models handle sample size differences among study groups and uncertainties differently leading to slightly different results. While MCPs have consistently been found to underestimate isotope space size and overlap [83] and are not calculated here, both SEA and KUD models are presented. Each of these models can be customized by using a pre-determined percentage of the test data, known as a contour level (or interval). Contour levels can be set at any percentage, but here we generate isotope space measures at two common levels: 75% and 95%. The purpose of applying a contour level is to prevent outliers or other extreme datapoints from overly influencing estimates of isotope space size or overlap. Best practices call for calculating isotope space size at more than one contour level to assess the stability of model measurements and to identify how outliers may affect isotope space measures [83,84].

In standard ellipse area (SEA) models, ellipses are constructed by calculating radii based on the test data defined by the contour level indicated. SEA models have lower sensitivity to sample size differences among study groups and minimize the effect of outliers but will always be in the shape of an ellipse that may include unused or exclude used areas of isotope space. Elliptical models also assume that isotope data are independent and normally distributed in multivariate space [82], although archaeological isotope datasets are known or often suspected to be prone to non-normality [85]. Kernel utilization density (KUD) models are generated by summing two kernel functions over observed datapoints with the total area defined as the minimum size that includes all datapoints within the contour level under consideration free of distributional assumptions or pre-set grid shapes, such as ellipses [83]. Both SEA and KUD models calculate isotope space overlap as the size of the overlapping region between the isotope space area size of group A and the isotope space area size of group B divided by the total isotope space area of group B (and the inverse to get a measure of how much group B overlaps group A) [83]. The percentage of overlap is interpreted here as indicating how consistent the isotopic composition of the archaeological maize from Chincha (at least in terms of $\delta^{15}N$ and $\delta^{34}S$) is with any of the experimental fertilizer conditions. While the specifics of the research question and nature of the study groups would determine what percentages of overlap are consequential, overlap of >60% is generally considered to be quite high and indicative of similar values [84].

Descriptive and inferential statistics were conducted in R (ver. 4.2.3). We applied an *a priori* significance level of $\alpha = 0.05$ for all statistical tests.

## Supporting information

**S1 File. Supplementary text, figures, tables, and references supporting the study.** Text A. Avian zooarchaeology on the Peruvian southern coast. Text B. Maize isotope physiology. S1 Fig. Maize cobs from UC-008 Tomb 1 in the middle Chincha Valley, Peru. S2 Fig. Architectural friezes from major administrative sites in the Chincha and Pisco valleys depicting seabirds, fish, and possible sprouting maize. S3 Fig. Bivariate plots of wt% N and C/N atomic ratio vs. $\delta^{15}N$ values of all archaeological maize from Chincha analyzed in this study, and bivariate plots of wt% N and C/N atomic ratio vs. $\delta^{15}N$ values of archaeological maize from Chincha analyzed at a target weight of 1.0 mg and used for all statistical analyses. S4 Fig. Bivariate plots of $\delta^{34}S$ vs. C:S ratios and $\delta^{34}S$ vs. N:S ratios of all archaeological maize from Chincha analyzed in this study. S1 Table. Avifauna results from Jahuay, Cerro Azul and Lo Demás. S2 Table. Summary statistics for Chincha maize and comparative sample from Chile. S3 Table. Comparison of estimates of isotopic space size from standard ellipse area (SEA) and kernel utilization density (KUD). S4 Table. Pair-wise isotopic space overlaps from standard ellipse area (SEA) and kernel utilization density (KUD).
(DOCX)

**S1 Dataset. Twenty radiocarbon dates from eleven graves sampled for maize.**
(XLSX)

**S2 Dataset. All isotopic data and standards.**
(XLSX)

## Acknowledgments

All necessary permits were obtained for the described study, which complied with all relevant regulations. The authors thank the Peruvian Ministry of Culture for granting us permits (206-2013-DGPC-VMPCIC/MC, 218-2015-DGPA-VMPCIC/MC, 107-2016-VMPCIC-MC, 145-2017-DGPA-VMPCIC/MC, and 148-2018-DGPA-VMPCIC/MC) to conduct this study. We appreciate the support from the Institute of Field Research, the Cotsen Institute of Archaeology, and the Archaeology Program at Boston University. Charles Stanish and Henry Tantaleán co-directed the Programa Arqueológico Chincha (PACH), which was essential for this research. Alexis Rodríguez, Richard Espino, Irving Aragonéz, and R. William Espino, along

with the students and staff of the Chincha Archaeological Field School and the Proyecto de Investigación Arqueológica de Jahuay, made important contributions throughout the research process. For maize samples, Sabrina Nino and Faith Evans (UC Merced) assisted with sample preparation in the Skeletal & Environmental Isotope Laboratory, Dr. Robin Trayler and the staff of the Stable Isotope Ecosystems Laboratory at UC Merced performed carbon and nitrogen isotope and elemental analysis, and the UC Davis Stable Isotope Facility performed sulfur isotope analysis. We are grateful to Hervé Bocherens, Valentina García-Huidobro, and Peter Tung in the Biologeology Working Group at the University of Tübingen, Germany; and thank Viorel Atudorei and Seth Newsome for feedback on plant quality control measures, aridity mechanisms, and plant diagenesis. Thanks to Erik Marsh for recalculating the ΔR value used for this paper.

## Author contributions

**Conceptualization:** Jacob L. Bongers, Emily B. P. Milton, Jo Osborn, Joshua R. Robinson, Beth K. Scaffidi.

**Investigation:** Jacob L. Bongers, Emily B. P. Milton, Jo Osborn, Beth K. Scaffidi.

**Methodology:** Jacob L. Bongers, Emily B. P. Milton, Jo Osborn, Dorothée G. Drucker, Joshua R. Robinson, Beth K. Scaffidi.

**Visualization:** Jacob L. Bongers, Emily B. P. Milton, Jo Osborn, Joshua R. Robinson.

**Writing – original draft:** Jacob L. Bongers, Emily B. P. Milton, Jo Osborn, Joshua R. Robinson, Beth K. Scaffidi.

**Writing – review & editing:** Jacob L. Bongers, Emily B. P. Milton, Jo Osborn, Dorothée G. Drucker, Joshua R. Robinson, Beth K. Scaffidi.

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
