## [Decision Letter · Decision Letter 0]

27 Jun 2025

PONE-D-25-25769Seabirds shaped the expansion of pre-Inca society in PeruPLOS ONE

Dear Dr. Bongers,

Thank you for submitting your manuscript to PLOS ONE. After careful consideration, we feel that it has merit but does not fully meet PLOS ONE’s publication criteria as it currently stands. Therefore, we invite you to submit a revised version of the manuscript that addresses the points raised during the review process.

We look forward to receiving your revised manuscript.

Kind regards,

Simon Belle, Ph.D.

Academic Editor

PLOS ONE

2. In your manuscript, please provide additional information regarding the specimens used in your study. Ensure that you have reported human remain specimen numbers and complete repository information, including museum name and geographic location.

For more information on PLOS ONE's requirements for paleontology and archeology research, see https://journals.plos.org/plosone/s/submission-guidelines#loc-paleontology-and-archaeology-research .

Additional Editor Comments:

Dear Authors,

Your manuscript PONE-D-25-25769 "Seabirds shaped the expansion of pre-Inca society in Peru" has now been seen by two external reviewers whose comments are listed at the end of these lines.

The reviews were generally favorable, but both reviewers provided some suggestions of edits that could make the manuscript stronger.

When preparing your revised manuscript, you are asked to carefully consider the reviewer comments which are attached, and submit a list of responses to the comments.

We look forward to receiving a new version of your manuscript.

With kind regards,

Reviewers' comments:

Reviewer's Responses to Questions

**Comments to the Author**

1. Is the manuscript technically sound, and do the data support the conclusions?

Reviewer #1: Yes

Reviewer #2: Yes

2. Has the statistical analysis been performed appropriately and rigorously? 

Reviewer #1: Yes

Reviewer #2: Yes

3. Have the authors made all data underlying the findings in their manuscript fully available?

Reviewer #1: Yes

Reviewer #2: Yes

4. Is the manuscript presented in an intelligible fashion and written in standard English?

Reviewer #1: Yes

Reviewer #2: Yes

5. Review Comments to the Author

Reviewer #1: The paper by Bongers et al presents new and original data about the possible fertilization of maize using seabird guano in the Chincha Valley (Peru) from the Late Formative until the Colonial Period.

Even though this case seems to be a good example on the use of fertilizers in the Andean region, I am worried about certain methodological aspects that debilitate the study:

- First of all, it is not clear to me why the authors analyzed surface-collected maize and not maize from their current or previous excavations in the cemeteries. The permanent exposure to sunlight, wind, dust, etc., could clearly affect the preservation of the archaeobotanical material. This is a very arid region and sun can affect negatively the maize conservation, cracking the cobs and allowing for contamination.

- Usually when analyzing archaeological plants for stable isotope analysis is necessary to run separately carbon and nitrogen isotopes. This, because the amount of nitrogen in plants is considerable low and the nitrogen peaks in the IRMS will not be high enough, generating confusing data. In fact, after doing the carbon isotope run, it is possible to calculate the amount in mg that will be needed for the nitrogen run, making sure to obtain a decent nitrogen peak and good d15N values. By reading the materials and methods section of the paper, it seems that the authors did just one run (with duplicates and triplicates) for d13C and d15N. For me, this is complicated in terms of methodology, as I find it difficult to trust the data. Moreover, the amount of maize measured in the IRMS, approx 1 mg, is really low compared to what you would expect for an archaeobotanical dessicated plant (between 3-5 mg). In addition, maize is characterized by having a very low % of N, so it is compulsory to measure more sample (mg) to obtain good results. Would it be possible for the authors to re-analyze their samples by doing separate runs? This is of great relevance to give further support to their findings and interpretations.

- I do not understand why the authors analyze bird bone collagen. I think it is clear for the audience that these are marine birds and that they will definitely have high d15N values as they feed on fish (and fish from the Humboldt Current that will be even more enriched in 15N than other parts of the world). It is confusing and there is not a straight connection between the fertilizer findings with the zooarchaeological results. It doesn't make much sense to me. Actually, this data and its discussion seems a bit forced in the paper.

On a different note, I find the structure of the paper somehow disorganized, especially in the discussion section. The authors jump from archaeological aspects to more isotope-methodological ones, to then go back to the archaeology discussion. I think the paper will be much better if the archaeology and isotope-methodology sections were separated and not mixing up along the discussion.

I will suggest the authors to be a little more cautious on their interpretations about the use of seabird guano and its contribution to the development of the Chincha Kingdom. For sure this fertilizer had an enormous impact on the local populations. However, this study only considers 35 samples (surface collected). More samples are needed to make greater interpretations.

I honestly believe that by improving these aspects, the paper will make a stronger case study. I would like to highlight the importance of discussing the role of sulfur isotopes in crops, and what can they tell us about agricultural practices. Even though there is not much information about this in the literature, particularly in the Andes and dessicated crops, I congratulate the authors for exploring this new line of evidence.

Reviewer #2: The manuscript by Bongers et al. "Seabirds Shaped the Expansion of Pre-Inca Society in Peru" asks a clear question: did seabird guano play a critical role in the agricultural and sociopolitical development of the Andes, particularly in the Chincha Valley?

The study is well written, has a compelling topic, and is methodologically robust. Its multidisciplinary approach, with topics as environmental history, archaeology and economic anthropology, is commendable, and the integration of stable isotope analysis into a broader historical and social framework is carefully executed.

However, the manuscript exaggerates the novelty of his contribution. It does not adequately situate its findings within the broader group of Andean archaeological and paleoecological studies. The manuscript should present a more balanced view, moderating claims to originality and engaging more deeply with the existing literature, especially given the well-established research on guano use, irrigation, and maize economies along coastal Peru. Although the authors claim that their study provides “the strongest evidence to date” for pre-Inca guano fertilization, they fail to acknowledge previous foundational work—notably that of Szpak (2014), Makarewicz & Tuross (2012), and others—that has already demonstrated the isotopic impact of guano on maize and human diets in Peru and Chile.

To strengthen the manuscript, authors should incorporate key references that provide critical context:

Chepstow-Lusty et al. (2009), which addresses agriculture as a central economic activity for Inca and Pre-Inca societies.

Snyder (2011), who discusses innovative irrigation techniques and their role in food security, including the use of guano.

Tykot and others. (2006), which links the introduction of corn to rapid population growth.

Rodrigues and Micael (2021), who highlight the role of guano birds in the expansion and prosperity of the Inca Empire and previous cultures.

Other concerns include:

Although guano is emphasized, alternative nitrogen sources such as fish-based composts or legume rotation are not adequately considered. Addressing these alternatives is important to validate the distinct isotopic signal of guano.

Although the discussion acknowledges the variability and limited understanding of sulfur uptake in plants, it still relies on δ³⁴S values to infer marine influence in paleodietary contexts. A stronger case would include experimental control data, such as fertilization trials under old analog conditions.

The manuscript does not sufficiently clarify the chronological precision or geographic specificity of the corn and guano samples. Without more precise dating, it is difficult to determine when guano fertilization became common in the Chincha Valley.

Finally, the discussion would benefit from exploring why guano fertilization was apparently adopted in the Chincha Valley but not uniformly in neighboring regions, and how adoption patterns may have varied socially or geographically.

6. PLOS authors have the option to publish the peer review history of their article (what does this mean? ). If published, this will include your full peer review and any attached files.

**Do you want your identity to be public for this peer review?** For information about this choice, including consent withdrawal, please see our Privacy Policy .

Reviewer #1: No

Reviewer #2: No

---

## [Author Response · Author response to Decision Letter 1]

29 Jul 2025

We appreciate the insightful feedback from the reviewers, which has improved the paper.

Reviewer #1:

The paper by Bongers et al presents new and original data about the possible fertilization of maize using seabird guano in the Chincha Valley (Peru) from the Late Formative until the Colonial Period. Even though this case seems to be a good example on the use of fertilizers in the Andean region, I am worried about certain methodological aspects that debilitate the study:

First of all, it is not clear to me why the authors analyzed surface-collected maize and not maize from their current or previous excavations in the cemeteries. The permanent exposure to sunlight, wind, dust, etc., could clearly affect the preservation of the archaeobotanical material. This is a very arid region and sun can affect negatively the maize conservation, cracking the cobs and allowing for contamination.

We understand the reviewer’s concern. All organic archaeological materials, regardless of depositional context, have the potential to undergo contamination after deposition. Typically, this is addressed via quality control screening measures. While quality control assessments for plant isotopic values remain an ongoing area of research, we have done our best to address diagenetic concerns throughout our methodology. We sampled from whole, visually intact maize cobs (no visible cracks), avoiding areas that appeared damaged or particularly dirty. We then applied the recommended cleaning protocol [1] for nitrate contaminants. We discuss these protocols in more detail in our main text and supplemental materials (lines 560-569; supplemental figure S3).

Usually when analyzing archaeological plants for stable isotope analysis is necessary to run separately carbon and nitrogen isotopes. This, because the amount of nitrogen in plants is considerable low and the nitrogen peaks in the IRMS will not be high enough, generating confusing data. In fact, after doing the carbon isotope run, it is possible to calculate the amount in mg that will be needed for the nitrogen run, making sure to obtain a decent nitrogen peak and good d15N values. By reading the materials and methods section of the paper, it seems that the authors did just one run (with duplicates and triplicates) for d13C and d15N. For me, this is complicated in terms of methodology, as I find it difficult to trust the data. Moreover, the amount of maize measured in the IRMS, approx 1 mg, is really low compared to what you would expect for an archaeobotanical dessicated plant (between 3-5 mg). In addition, maize is characterized by having a very low % of N, so it is compulsory to measure more sample (mg) to obtain good results. Would it be possible for the authors to re-analyze their samples by doing separate runs? This is of great relevance to give further support to their findings and interpretations.

We appreciate the reviewer’s concerns; however, we suggest that a re-analysis at higher masses is not presently substantiated. We applied published protocols from leading regional plant studies, such as Szpak and Chiou 2019 [2], who suggested a minimum threshold of 0.5 mg as sufficient for analysis. Given the differences in machinery and protocols among labs, we also relied on the expertise of the lab specialists handling our samples to ensure materials were prepared appropriately for analysis. For example, in our initial run, several samples weight at ~0.5 mg samples did not produce adequate peak areas (aligned with reviewer concerns). Following a consultation with the U.C. Merced lab director, these samples were re-run at higher masses (>1mg) with good results (see attached reports EA20231220_report_0.5mg and EA20240207_report_1.0mg, which detail this occurrence, as well as lab linearity corrections and peak information for our samples).

All samples analysed at ≥1 mg fell within accepted ranges for the machine sensitivity–with good peak areas and no apparent linearity effects–suggesting our masses were adequate for analysis. Prompted by the reviewer’s concerns, we consulted with additional lab personnel, including folk at the University of New Mexico Center for Stable Isotopes, regarding machine settings and appropriate masses for plant analysis. We were informed by both labs that only plants with very high C:N ratios, such as those typical of wood (200-1000), require independent measurements for nitrogen. Our sample C:N ratios do not meet this threshold. We were also informed that studies focused on nitrogen fixation, or like analyses requiring highly precise and accurate results, can request to analyse plants with lower C:N ratios in nitrogen-only mode; however, our research questions do not require this setting, as even a few per mille of inaccuracy would not change our interpretations. As sample mass appears to be an emerging area of concern for archaeologists, we will request our reviews are made open to provide full transparency to readers on our methodological decision making.

I do not understand why the authors analyze bird bone collagen. I think it is clear for the audience that these are marine birds and that they will definitely have high d15N values as they feed on fish (and fish from the Humboldt Current that will be even more enriched in 15N than other parts of the world). It is confusing and there is not a straight connection between the fertilizer findings with the zooarchaeological results. It doesn't make much sense to me. Actually, this data and its discussion seems a bit forced in the paper.

The bird bones provide a baseline for isotopic expectations, as the isotopic composition of marine and terrestrial ecosystems can vary significantly, even over short distances. As the reviewer notes, the highly productive Humboldt Current and associated marine ecosystem services are unique to the coast of South America; therefore, it is essential to establish isotopic expectations for the region and period, as we note on lines 114-115 and 207-210. As there are few d15N / d13C data, and almost no d34S data available for archaeological marine fauna in the Chincha area (or even Peru), we measured carbon, nitrogen, and sulfur isotopes from various bird bones as a means of establishing a baseline to estimate local isotopic variation within the guano system. We have added clarification that this was the reason for our study design on lines 370-372. We emphasize that our data reinforce the need for local baseline construction, as our sulfur values were ~5‰ lower than the generally accepted +20‰ cited for global reservoirs (see lines 376-378). As such, our data have helped establish a lower threshold for possible marine sulfur values.

On a different note, I find the structure of the paper somehow disorganized, especially in the discussion section. The authors jump from archaeological aspects to more isotope-methodological ones, to then go back to the archaeology discussion. I think the paper will be much better if the archaeology and isotope-methodology sections were separated and not mixing up along the discussion.

Thank you for this constructive recommendation. We have revised the Discussion section to improve clarity and structure. We now begin the section with a brief introduction outlining its focus on both methodological and empirical contributions. We have also separated the methodological and archaeological discussions into distinct subsections to enhance readability and ensure a more coherent flow of ideas.

I will suggest the authors to be a little more cautious on their interpretations about the use of seabird guano and its contribution to the development of the Chincha Kingdom. For sure this fertilizer had an enormous impact on the local populations. However, this study only considers 35 samples (surface collected). More samples are needed to make greater interpretations.

We are grateful to the reviewer for this important and thoughtful comment. While our results offer high-probability, multidisciplinary evidence for the use of seabird guano as a pre-Hispanic agricultural amendment, we agree that our sample size necessitates caution in drawing broader conclusions about the scale and sociopolitical impact of this practice within the Chincha Kingdom.

We have tempered our interpretations regarding the role of guano in the emergence of the Chincha polity. For example, in the Abstract on lines 47-48, we now, “suggest that seabird guano fertilization played an important role in the sociopolitical and economic expansion of the Chincha Kingdom.” In the Discussion, we qualify our interpretation by stating that guano, “likely constituted a critical source of economic and political power,” and, “may have played a key role” in local negotiations with the Inca Empire (lines 492, 512-513).

I honestly believe that by improving these aspects, the paper will make a stronger case study. I would like to highlight the importance of discussing the role of sulfur isotopes in crops, and what can they tell us about agricultural practices. Even though there is not much information about this in the literature, particularly in the Andes and dessicated crops, I congratulate the authors for exploring this new line of evidence.

It’s been a unique [challenging] experience to collect and situate new sulfur data within the regional context, so we sincerely appreciate the reviewer’s recognition of this component :)

Reviewer #2:

The manuscript by Bongers et al. "Seabirds Shaped the Expansion of Pre-Inca Society in Peru" asks a clear question: did seabird guano play a critical role in the agricultural and sociopolitical development of the Andes, particularly in the Chincha Valley?

The study is well written, has a compelling topic, and is methodologically robust. Its multidisciplinary approach, with topics as environmental history, archaeology and economic anthropology, is commendable, and the integration of stable isotope analysis into a broader historical and social framework is carefully executed. However, the manuscript exaggerates the novelty of his contribution. It does not adequately situate its findings within the broader group of Andean archaeological and paleoecological studies. The manuscript should present a more balanced view, moderating claims to originality and engaging more deeply with the existing literature, especially given the well-established research on guano use, irrigation, and maize economies along coastal Peru. Although the authors claim that their study provides “the strongest evidence to date” for pre-Inca guano fertilization, they fail to acknowledge previous foundational work—notably that of Szpak (2014), Makarewicz & Tuross (2012), and others—that has already demonstrated the isotopic impact of guano on maize and human diets in Peru and Chile.

We appreciate the author’s perspective on the impact of our contributions, and have sought to moderate our claims, as we address to reviewer 1 above, and further, below. We also thank you for directing us to these references; Szpak’s graduate and subsequent work has been instrumental to development of isotopic fertilization studies in the region. We’ve added acknowledgement of these studies, which can be found on lines: 57-58, 100, 104

To strengthen the manuscript, authors should incorporate key references that provide critical context:

Chepstow-Lusty et al. (2009), which addresses agriculture as a central economic activity for Inca and Pre-Inca societies.

Snyder (2011), who discusses innovative irrigation techniques and their role in food security, including the use of guano.

Tykot and others. (2006), which links the introduction of corn to rapid population growth.

Rodrigues and Micael (2021), who highlight the role of guano birds in the expansion and prosperity of the Inca Empire and previous cultures.

We appreciate the reviewer’s concerns regarding the novelty of our argument and thank them for their provision of additional papers on this topic. We acknowledge (and cite) several studies that have conducted a multidisciplinary investigation of fertilization and have now sought to moderate our claims throughout the article. We feel our research is novel because we provide some of the first stable sulfur data for archaeological maize and guano birds in the Americas (previous studies have only contributed small (n=1) sample sizes with no quality control information). However, our argument about guano use does not rely solely on isotopic evidence, and we suggest our insights are unique for the south coast region; we consider geography (proximity to some of the most abundant guano deposits in the Andean region on the Chincha Islands) as well as archaeological (e.g., iconography, zooarchaeological remains, etc.) and historical (colonial-era documents) data reported in our paper. We agree with the reviewer that the paper would benefit from deeper engagement with the existing literature and are grateful for the thoughtful list of relevant sources on guano use and agricultural strategies. We have now cited all sources in the manuscript.

Although guano is emphasized, alternative nitrogen sources such as fish-based composts or legume rotation are not adequately considered. Addressing these alternatives is important to validate the distinct isotopic signal of guano.

We considered the possibility of other fertilizers but see this was not clear in our original text. As such, we’ve added a statement considering other fertilizers for our samples <20‰ (239-240). To date, guano is the only known amendment to produce >20‰ nitrogen values in agricultural crops (see lines 97-105 for more detail) (Szpak 2012a, 2012b, and 2014) [3–5], therefore, we still consider seabird guano as the only parsimonious explanation for many of our samples. Although the discussion acknowledges the variability and limited understanding of sulfur uptake in plants, it still relies on δ³⁴S values to infer marine influence in paleodietary contexts. A stronger case would include experimental control data, such as fertilization trials under old analog conditions.

We agree on the need for additional analog trials; however, we suggest there is some urgency in publishing our current dataset, as is, and allowing experimental studies to follow. Stable sulfur is poised to become a new trend in the region, and as we outline repeatedly throughout the paper, should not be used uncritically (and merits more foundational work!).

Recent archaeological fertilization studies have increasingly noted a potential for "contamination” of terrestrial food systems (at the human and animal levels) by marine fertilizers; the implications of such contamination for paleodietary reconstructions of past agricultural communities are significant. Whether marine δ³⁴S is reliably routed into plants via fertilization is at the center of this discussion––as the common, two-isotope analyses of carbon and nitrogen have been implicated by the well-documented enrichment of 15N in marine-fertilized C4 plants.

If δ³⁴S values consistently increase to marine levels (≥+15‰) in terrestrial plants, we cannot reliably use δ³⁴S to discriminate between marine and terrestrial inputs in fertilizer-using periods. If, however, marine-fertilized plants do not consistently enrich in 34S––as existing experimental studies and our newly presented maize data suggest––then δ³⁴S may remain a reliable, and relatively simple/accessible method for better untangling human/animal foodwebs on the coast.

It would be a shame to throw the baby (δ³⁴S) out with the bathwater! As such, we’ve used our data to initiate this exact discussion, describing the complexity of the problem and emphasizing the criticality of more mechanistic studies on sulfur (lines 415-417; 430-432). Using our data and interdisciplinary interpretations, we hope to help direct people towards, dare we say, “fertile” areas of future research.

The manuscript does not sufficiently clarify the chronological precision or geographic specificity of the corn and guano samples. Without more precise dating, it is difficult to determine when guano fertilization became common in the Chincha Valley.

Thank you for requesting clarification regarding the chronology of our samples. As noted in the “Maize carbon and nitrogen isotopes” section (lines 225-226), u

---

## [Decision Letter · Decision Letter 1]

18 Sep 2025

PONE-D-25-25769R1Seabirds shaped the expansion of pre-Inca society in PeruPLOS ONE

Dear Dr. Bongers,

Thank you for submitting your manuscript to PLOS ONE. After careful consideration, we feel that it has merit but does not fully meet PLOS ONE’s publication criteria as it currently stands. Therefore, we invite you to submit a revised version of the manuscript that addresses the points raised during the review process.

**ACADEMIC EDITOR:** 

Overall, the reviewer has expressed a positive view of your work, acknowledging the improvements you have made since the initial submission.

However, the reviewer has also highlighted a few significant issues that need to be carefully addressed in order to enhance the quality of the paper. These concerns, although distinct from the points raised in the first round of review, suggest that there is still room for further improvement. Addressing these points effectively will likely strengthen your manuscript and increase its chances of successful publication.

We look forward to receiving your revised manuscript.

Kind regards,

Simon Belle, Ph.D.

Academic Editor

PLOS ONE

Journal Requirements:

Additional Editor Comments (if provided):

Reviewer #3:

The authors present a study that complements existing knowledge on the use of fertilizers by pre-Hispanic populations. In this regard, the paper’s contribution lies in its focus on a specific area—namely, the Chincha Valley—and its discussion of how fertilization may have supported coastal expansion models.

How ever some improvements to the manuscripts:

When the authors refer to Chile, they must specify that it corresponds to the Tarapacá region. Chile is a country over 5,000 km long, with diverse ecological niches, landscapes, and pre-Hispanic cultures. Therefore, to ensure accuracy, it is requested that references to Chile be specified at least in the introduction and background sections as “Tarapacá, Northern Chile” (lines 44, 66, 117, and 272). The archaeological sites of Tarapacá correspond to various locations from the Late Formative, Late Intermediate Period (LIP), and Late Horizon (LH) in Tarapacá.

The research question clearly addresses the cultural significance and impact of guano use in pre-Hispanic societies.

The map should be improved in terms of geographic contextualization. While it shows the spatial relationship between the Chincha Valley, the Tarapacá region, and the sampled sites, it must be enhanced by including information such as a relevant capital city or by adding latitude and longitude coordinates. Additionally, it is requested that the icons representing sampling zones be made clearer and more contrasting.

Given that the sample size is small, clarification is needed regarding the 35 maize cobs mentioned. In Supplementary Figure 1 only fragments of cobs are shown, so it must be clarified whether the 35 cobs refer to complete specimens or fragments. If they are fragments, the authors should explain how the minimum number of 35 specimens was determined, since multiple fragments may belong to the same cob, reducing the actual number of individual samples and the sampling universe. This is essential to clarify due to the statistical analyses being conducted.

I strongly recommend to include some of the morphological characteristics of the archaeological maize samples, especially since not all of them fall within the same range of isotope values

Results:

Regarding the material culture findings, the idea is presented that bird iconography on various media may link marine fertility to agricultural productivity. This concept aligns well with Andean logics of duality, complementarity, and reciprocity, but it is not fully developed. It is suggested that the authors expand on this interpretation and revisit the research question by exploring the relationship between coastal resources and agriculture—especially if the aim is to show how seabirds contributed to the development and expansion of human groups in Chincha.

It is unclear why the maize cobs were not directly dated. Direct dating of cobs is consistent with the chronology of various Andean archaeological sites, including those in Tarapacá, Northern Chile, referenced in the manuscript (see Vidal et al. 2021, Vidal et al. 2024). The claim that maize cobs may yield misleading radiocarbon results lacks support. Grobman et al 2012´s observation is a specific inconsistency in the Huaca Prieta context and should not be generalized. This point must be clarified, and ideally, the cobs should be directly dated.

Although the δ34S results are inconclusive, their inclusion is appreciated and valuable as comparative data for future analyses. This new type of experimental approach is appreciated.

Figures 4, 5, and 6: These figures have low resolution and should be improved to allow proper reading and interpretation.

Discusion:

The relationship between the use of seabird guano as fertilizer and the development of the Chincha Kingdom must be further explored. The manuscript should explore in greater depth how agriculture enhanced by seabird guano fertilization may have contributed to the emergence of complexity in the Chincha Kingdom. For example, was agriculture the foundation for new systems of cooperation or competition? Did it lead to new forms of territorial control? Since this element is central to the research question, it should be critically evaluated, considering relevant archaeological social theory—such as the work of Stanish. The use of the Maritime Foundations of Andean Civilization framework, while appropriate, could be enriched by incorporating more updated perspectives. While the coast is ecologically significant, it is also shaped by human decisions and multiples factors.

Reviewers' comments:

Reviewer's Responses to Questions

**Comments to the Author**

1. If the authors have adequately addressed your comments raised in a previous round of review and you feel that this manuscript is now acceptable for publication, you may indicate that here to bypass the “Comments to the Author” section, enter your conflict of interest statement in the “Confidential to Editor” section, and submit your "Accept" recommendation.

Reviewer #3: (No Response)

2. Is the manuscript technically sound, and do the data support the conclusions?

Reviewer #3: Partly

3. Has the statistical analysis been performed appropriately and rigorously? 

Reviewer #3: I Don't Know

4. Have the authors made all data underlying the findings in their manuscript fully available?

Reviewer #3: Yes

5. Is the manuscript presented in an intelligible fashion and written in standard English?

Reviewer #3: Yes

6. Review Comments to the Author

Reviewer #3: The authors present a study that complements existing knowledge on the use of fertilizers by pre-Hispanic populations. In this regard, the paper’s contribution lies in its focus on a specific area—namely, the Chincha Valley—and its discussion of how fertilization may have supported coastal expansion models.

How ever some improvements to the manuscripts:

When the authors refer to Chile, they must specify that it corresponds to the Tarapacá region. Chile is a country over 5,000 km long, with diverse ecological niches, landscapes, and pre-Hispanic cultures. Therefore, to ensure accuracy, it is requested that references to Chile be specified at least in the introduction and background sections as “Tarapacá, Northern Chile” (lines 44, 66, 117, and 272). The archaeological sites of Tarapacá correspond to various locations from the Late Formative, Late Intermediate Period (LIP), and Late Horizon (LH) in Tarapacá.

The research question clearly addresses the cultural significance and impact of guano use in pre-Hispanic societies.

The map should be improved in terms of geographic contextualization. While it shows the spatial relationship between the Chincha Valley, the Tarapacá region, and the sampled sites, it must be enhanced by including information such as a relevant capital city or by adding latitude and longitude coordinates. Additionally, it is requested that the icons representing sampling zones be made clearer and more contrasting.

Given that the sample size is small, clarification is needed regarding the 35 maize cobs mentioned. In Supplementary Figure 1 only fragments of cobs are shown, so it must be clarified whether the 35 cobs refer to complete specimens or fragments. If they are fragments, the authors should explain how the minimum number of 35 specimens was determined, since multiple fragments may belong to the same cob, reducing the actual number of individual samples and the sampling universe. This is essential to clarify due to the statistical analyses being conducted.

I strongly recommend to include some of the morphological characteristics of the archaeological maize samples, especially since not all of them fall within the same range of isotope values

Results:

Regarding the material culture findings, the idea is presented that bird iconography on various media may link marine fertility to agricultural productivity. This concept aligns well with Andean logics of duality, complementarity, and reciprocity, but it is not fully developed. It is suggested that the authors expand on this interpretation and revisit the research question by exploring the relationship between coastal resources and agriculture—especially if the aim is to show how seabirds contributed to the development and expansion of human groups in Chincha.

It is unclear why the maize cobs were not directly dated. Direct dating of cobs is consistent with the chronology of various Andean archaeological sites, including those in Tarapacá, Northern Chile, referenced in the manuscript (see Vidal et al. 2021, Vidal et al. 2024). The claim that maize cobs may yield misleading radiocarbon results lacks support. Grobman et al 2012´s observation is a specific inconsistency in the Huaca Prieta context and should not be generalized. This point must be clarified, and ideally, the cobs should be directly dated.

Although the δ34S results are inconclusive, their inclusion is appreciated and valuable as comparative data for future analyses. This new type of experimental approach is appreciated.

Figures 4, 5, and 6: These figures have low resolution and should be improved to allow proper reading and interpretation.

Discusion:

The relationship between the use of seabird guano as fertilizer and the development of the Chincha Kingdom must be further explored. The manuscript should explore in greater depth how agriculture enhanced by seabird guano fertilization may have contributed to the emergence of complexity in the Chincha Kingdom. For example, was agriculture the foundation for new systems of cooperation or competition? Did it lead to new forms of territorial control? Since this element is central to the research question, it should be critically evaluated, considering relevant archaeological social theory—such as the work of Stanish. The use of the Maritime Foundations of Andean Civilization framework, while appropriate, could be enriched by incorporating more updated perspectives. While the coast is ecologically significant, it is also shaped by human decisions and multiples factors.

7. PLOS authors have the option to publish the peer review history of their article (what does this mean? ). If published, this will include your full peer review and any attached files.

**Do you want your identity to be public for this peer review?** For information about this choice, including consent withdrawal, please see our Privacy Policy .

Reviewer #3: No

---

## [Author Response · Author response to Decision Letter 2]

22 Oct 2025

Responses to Feedback

We appreciate the insightful feedback from this reviewer, which has improved the paper.

Reviewer #3:

The authors present a study that complements existing knowledge on the use of fertilizers by pre-Hispanic populations. In this regard, the paper’s contribution lies in its focus on a specific area—namely, the Chincha Valley—and its discussion of how fertilization may have supported coastal expansion models.

However some improvements to the manuscripts:

When the authors refer to Chile, they must specify that it corresponds to the Tarapacá region. Chile is a country over 5,000 km long, with diverse ecological niches, landscapes, and pre-Hispanic cultures. Therefore, to ensure accuracy, it is requested that references to Chile be specified at least in the introduction and background sections as “Tarapacá, Northern Chile” (lines 44, 66, 117, and 272). The archaeological sites of Tarapacá correspond to various locations from the Late Formative, Late Intermediate Period (LIP), and Late Horizon (LH) in Tarapacá.

Thank you for this point. We have amended the text so that the first mention of the Chilean data now states it is from Tarapacá.

The research question clearly addresses the cultural significance and impact of guano use in pre-Hispanic societies.

We appreciate this assessment!

The map should be improved in terms of geographic contextualization. While it shows the spatial relationship between the Chincha Valley, the Tarapacá region, and the sampled sites, it must be enhanced by including information such as a relevant capital city or by adding latitude and longitude coordinates. Additionally, it is requested that the icons representing sampling zones be made clearer and more contrasting.

We have updated the map to include the capital city of Lima and to improve the contrast of the icons.

Given that the sample size is small, clarification is needed regarding the 35 maize cobs mentioned. In Supplementary Figure 1 only fragments of cobs are shown, so it must be clarified whether the 35 cobs refer to complete specimens or fragments. If they are fragments, the authors should explain how the minimum number of 35 specimens was determined, since multiple fragments may belong to the same cob, reducing the actual number of individual samples and the sampling universe. This is essential to clarify due to the statistical analyses being conducted.

I strongly recommend to include some of the morphological characteristics of the archaeological maize samples, especially since not all of them fall within the same range of isotope values

We have updated the text to clarify (1) the maize cobs are all more than 50% complete (228) and that (2) MNI may be established as the maize fragments were collected from 26 distinct tombs across 14 cemeteries (227-228). This mitigates the issue that the reviewer raises.

We appreciate the reviewer’s suggestion to include morphological data for the maize samples.

Morphological measurements were not collected for these specimens, although we are considering including them in future studies. We agree that a detailed morphological analysis would be valuable future research. However, the focus for this study is to evaluate isotopic evidence for fertilization practices rather than to characterize the morphology of individual maize samples.

Results:

Regarding the material culture findings, the idea is presented that bird iconography on various media may link marine fertility to agricultural productivity. This concept aligns well with Andean logics of duality, complementarity, and reciprocity, but it is not fully developed. It is suggested that the authors expand on this interpretation and revisit the research question by exploring the relationship between coastal resources and agriculture—especially if the aim is to show how seabirds contributed to the development and expansion of human groups in Chincha.

We agree with the reviewer that, ideally, we would expand the ideological aspects of guano fertilization and its relationship to seabirds in iconography and ethnohistory. However, due to the multidisciplinary approach of this article, and the extensive discussion required for stable isotopic data, we have determined that this is beyond the scope of this article. We have future work planned and would like to include these ideas!

It is unclear why the maize cobs were not directly dated. Direct dating of cobs is consistent with the chronology of various Andean archaeological sites, including those in Tarapacá, Northern Chile, referenced in the manuscript (see Vidal et al. 2021, Vidal et al. 2024). The claim that maize cobs may yield misleading radiocarbon results lacks support. Grobman et al 2012´s observation is a specific inconsistency in the Huaca Prieta context and should not be generalized. This point must be clarified, and ideally, the cobs should be directly dated.

Our maize cobs are dated as best as possible given (1) our available resources and (2) what is presently known regarding radiocarbon dating of uncharred material. We suggest Grobman et al. (2012)’s study indicates caution should be taken when dating uncharred maize cobs, as we have in our sample. While Grobman et al. 2012 found that charred cobs produced radiocarbon results consistent with their stratigraphic context, all nine uncharred cobs, which represent six separate units, produced anomalously young dates, regardless of their stratigraphic positioning. We therefore contend that the issue is not specific to Huaca Prieta, but to the uncharred nature of the material. While we agree that an ideal scenario would involve direct dating of our samples, we do not feel that this is a justified use of resources at this time, as we cannot be sure those dates would provide reliable results––rather, Grobman et al. provide grounds to believe any dates on uncharred cob should be treated with caution. We realize we misstated that some of our cobs were partially charred and have updated the text to record their uncharred nature (556).

Although the δ34S results are inconclusive, their inclusion is appreciated and valuable as comparative data for future analyses. This new type of experimental approach is appreciated.

We appreciate the reviewer recognizing the value of these data to the wider study of maize fertilization.

Figures 4, 5, and 6: These figures have low resolution and should be improved to allow proper reading and interpretation.

We believe the issue has been caused by the file format provided to the reviewer but will ensure high-resolution images are provided for the final document.

Discussion:

The relationship between the use of seabird guano as fertilizer and the development of the Chincha Kingdom must be further explored. The manuscript should explore in greater depth how agriculture enhanced by seabird guano fertilization may have contributed to the emergence of complexity in the Chincha Kingdom. For example, was agriculture the foundation for new systems of cooperation or competition? Did it lead to new forms of territorial control? Since this element is central to the research question, it should be critically evaluated, considering relevant archaeological social theory—such as the work of Stanish. The use of the Maritime Foundations of Andean Civilization framework, while appropriate, could be enriched by incorporating more updated perspectives. While the coast is ecologically significant, it is also shaped by human decisions and multiples factors.

Thank you for raising these thoughtful points. To address how seabird guano fertilization contributed to the development of the Chincha Kingdom, we first must clarify the broader role that manuring can play in driving the expansion of large-scale societies. In the Introduction (lines 57-61), we note that sustainable agricultural practices can promote the expansion of societies by ensuring a stable food supply for growing populations. In the Discussion (lines 478-480), we have revised statements to emphasize that sustainable agricultural practices involving maize production can serve as an economic foundation for large-scale societies by providing reliable food resources that can sustain population growth. We have added the Lombardo et al. (2025) and Finucane (2009) citations to support our argument. Seabird guano fertilization is highlighted as one such sustainable practice, as it directly increases maize crop yields (lines 481-482). We draw attention to evidence of dense Late Intermediate Period and Late Horizon settlement and high population estimates for the late pre-Hispanic Chincha Valley, further demonstrating the potential effects that seabird guano fertilization can have on a large-scale polity such as the Chincha Kingdom (lines 481-484).

We provide a detailed discussion about how seabird guano contributed to cooperation and the emergence of complexity in the Chincha Kingdom (lines 493-512). For example, we write about how guano facilitated cooperation between Chincha Kingdom’s specialized farmers, fisherfolk, and merchants. We state, following Curatola’s (1997) model, that fisherfolk sailed to the nearby offshore islands to acquire guano and then subsequently provided it to farmers for enhancing maize production and to merchants for trading along the coast and the highlands, expanding not only Chincha’s agricultural productivity but also its mercantile influence (lines 501-504). This burgeoning network of specialists is a hallmark of the complexity of the Chincha Kingdom. We have added lines 504-507 to note that increased population levels and maize production arising from guano use may have intensified inter-group competition and new forms of territorial control, but these are hypotheses that need to be further explored archaeologically.

We agree that the coast is an ecologically important zone that is shaped by human decisions. To this point, we have stated that seabird guano fertilization may have represented an effective regional strategy for producing food despite the challenging, arid Andean coast (lines 454-461 and lines 487-492).

We feel we have incorporated Andean work sufficiently and provided a measured interpretation regarding the wider significance to the region while maintaining a manuscript of appropriate length. We feel that additional topics, such as competition, would be better addressed in future follow-up studies looking at wider regional patterns in similar data, and hope such a study will be possible soon. If the reviewer is working on similar research, we encourage them to reach out for a potential collaboration after the publication of this article.

---

## [Decision Letter · Decision Letter 2]

30 Dec 2025

PONE-D-25-25769R2Seabirds shaped the expansion of pre-Inca society in PeruPLOS One

Dear Dr. Bongers,

I am happy to inform you that I have secured an additional review for your manuscript.

The reviewer raised only a few minor issues, which should be straightforward to address.

Thank you very much for your patience throughout this process. I appreciate your cooperation, and I wish you a very Happy New Year.

We look forward to receiving your revised manuscript.

Kind regards,

Simon Belle, Ph.D.

Academic Editor

PLOS One

Journal Requirements:

Reviewers' comments:

Reviewer's Responses to Questions

**Comments to the Author**

1. If the authors have adequately addressed your comments raised in a previous round of review and you feel that this manuscript is now acceptable for publication, you may indicate that here to bypass the “Comments to the Author” section, enter your conflict of interest statement in the “Confidential to Editor” section, and submit your "Accept" recommendation.

Reviewer #4: (No Response)

2. Is the manuscript technically sound, and do the data support the conclusions?

Reviewer #4: Yes

3. Has the statistical analysis been performed appropriately and rigorously? 

Reviewer #4: Yes

4. Have the authors made all data underlying the findings in their manuscript fully available?

Reviewer #4: Yes

5. Is the manuscript presented in an intelligible fashion and written in standard English?

Reviewer #4: Yes

6. Review Comments to the Author

Reviewer #4: I should begin by noting that I was not one of the original reviewers for this manuscript, so I am reading it for the first time. As the manuscript has already gone through two review stages, the version I read was well-polished and addressed the previous reviewers’ concerns for the most part. Where a reviewer’s recommendations were not resolved, the authors made a satisfactory explanation as to why they chose not to make the revision (for example, Reviewer #3’s recommendation to radiocarbon date all maize samples, which I agree with the authors is not necessary for this study and given the contextual information already known for the samples).

I do not have any issue with the methodology, statistics, or conclusions drawn from the study, and believe it serves as an important contribution for archaeology in the Andean/coastal region. It is also relevant to other regions of the world, as the economics of fertilization is often ignored in archaeology. The new data regarding δ34S isotopes is interesting, even if it did not conform to expectations (“At present it remains unclear whether marine-fertilized terrestrial plants could become so enriched in 34S that they would enter marine isotope space”).

I reviewed the manuscript and supplemental data/document files. My comments are brief and should be easy to resolve. The first is the most significant, but the others are minor:

Figure 1 caption, line 119: “Map of Chincha study area, with middle valley cemeteries sampled for maize (marked by yellow squares)” The symbols for maize locations are red/pink circles. I think the yellow squares may have been on a previous version of the map.

Minor issues:

Line 110 typo: “however, associated field sprovided inconclusive results for”

Line 115 typo: “marine isotopic values iis lacking”

Line 128: “We conduct stable carbon, nitrogen, and sulfur analyses” add “isotopic” or “isotope” before “analyses”

Line 259 spacing typo: “there were no significant changes in δ34S values [47].In this study,”

Line 396: “terrestrial C4 crops” subscript 4

Line 490; “cooperation between” should be “among” since there are several groups listed, not two

Lines 555-556 typo: “The necessity and protocols for pre-treating desiccated and carbonized plants remains debated” add an “is” before “debated”

Line 638 swap period for comma: “into 8,5 x 5 millimeter”

Line 640 swap period for comma: “5,5 x 3,5 millimeter tin capsules”

Line 641 paragraph: be consistent with ‰ and spaces after numbers, as this paragraph has both versions (I would recommend checking the entire manuscript to be consistent)

Line 650 typo?: “n analytical error below 0.1 ‰, 0.2 ‰ and 0.4‰”

Lines 650-654: no superscript for isotopes on these lines

Line 665: “from middle valley” should this be capitalized?

Fig 2 caption: Since Plos One aims for a broad (if mostly academic) audience, should the common names of the birds be included too?

Fig 3 caption: “middle Chincha Valley” and “Middle Chincha Valley” both appear (this relates to the capitalization comment earlier)

Fig 6 caption: “C4 crops” subscript 4

Spacing for ± and numbers is inconsistent throughout the paper

7. PLOS authors have the option to publish the peer review history of their article (what does this mean? ). If published, this will include your full peer review and any attached files.

**Do you want your identity to be public for this peer review?** For information about this choice, including consent withdrawal, please see our Privacy Policy .

Reviewer #4: No

---

## [Author Response · Author response to Decision Letter 3]

2 Jan 2026

Responses to Feedback

We appreciate the insightful feedback from this reviewer, which has improved the paper.

Reviewer #4:

• Figure 1 caption, line 119: “Map of Chincha study area, with middle valley cemeteries sampled for maize (marked by yellow squares)” The symbols for maize locations are red/pink circles. I think the yellow squares may have been on a previous version of the map.

Thank you for pointing this out. We have amended the text to say that the cemeteries are marked by pink circles.

• Line 110 typo: “however, associated field sprovided inconclusive results for”

We have revised the text to say that “associated field studies provided inconclusive results for…”

• Line 115 typo: “marine isotopic values iis lacking”

• Line 128: “We conduct stable carbon, nitrogen, and sulfur analyses” add “isotopic” or “isotope” before “analyses”

• Line 259 spacing typo: “there were no significant changes in δ34S values [47].In this study,”

• Line 396: “terrestrial C4 crops” subscript 4

• Line 490; “cooperation between” should be “among” since there are several groups listed, not two

• Lines 555-556 typo: “The necessity and protocols for pre-treating desiccated and carbonized plants remains debated” add an “is” before “debated”

• Line 638 swap period for comma: “into 8,5 x 5 millimeter”

• Line 640 swap period for comma: “5,5 x 3,5 millimeter tin capsules”

We are grateful to you for catching these mistakes. They have all been addressed in our manuscript.

• Line 641 paragraph: be consistent with ‰ and spaces after numbers, as this paragraph has both versions (I would recommend checking the entire manuscript to be consistent)

Thanks for identifying this inconsistency. We are now consistent with ‰ and spaces after numbers.

• Line 650 typo?: “n analytical error below 0.1 ‰, 0.2 ‰ and 0.4‰”

Yes, this is a typo. We have removed ‘n’ from the sentence.

• Lines 650-654: no superscript for isotopes on these lines

• Line 665: “from middle valley” should this be capitalized?

• Fig 3 caption: “middle Chincha Valley” and “Middle Chincha Valley” both appear (this relates to the capitalization comment earlier)

We have included superscripts for isotopes on Lines 650-654. In our manuscript, we capitalize ‘Chincha Valley’ when used as part of the phrase ‘middle Chincha Valley;’ otherwise, ‘middle valley’ is not capitalized. This is now consistent in the manuscript.

• Fig 2 caption: Since Plos One aims for a broad (if mostly academic) audience, should the common names of the birds be included too?

This is an excellent point. We have now included the common names for the birds in the Figure 2 caption.

• Fig 6 caption: “C4 crops” subscript 4

• Spacing for ± and numbers is inconsistent throughout the paper

Thanks again for pointing out these issues. We have corrected the formatting of ‘C4’ in Figure 6 and ensured that spacing around the ± symbol and numerical values is consistent throughout the paper.

---

## [Editor Report · Decision Letter 3]

5 Jan 2026

Seabirds shaped the expansion of pre-Inca society in Peru

PONE-D-25-25769R3

Dear Dr. Bongers,

We’re pleased to inform you that your manuscript has been judged scientifically suitable for publication and will be formally accepted for publication once it meets all outstanding technical requirements.

Kind regards,

Simon Belle, Ph.D.

Academic Editor

PLOS One

---

## [Editor Report · Acceptance letter]

PONE-D-25-25769R3

PLOS One

Dear Dr. Bongers,

I'm pleased to inform you that your manuscript has been deemed suitable for publication in PLOS One. Congratulations! Your manuscript is now being handed over to our production team.

Kind regards,

on behalf of

Dr. Simon Belle

Academic Editor

PLOS One